# Short-range order in high entropy carbides

Shuguang Wei [1,6], Muhammad Waqas Qureshi[1,6], Jingrui Wei[1,6], Longfei Liu[1,6], Xuanxin Hu[1], Jianqi Xi[2], Siamak Attarian [1], Ranran Su [3], Hongliang Zhang [4], Evan Willing[5], Xudong Wang [1], Kumar Sridharan[5], Paul M. Voyles [1], John H. Perepezko [1] & Izabela Szlufarska [1] ✉

High-entropy carbides (HECs) are a new class of materials with properties that are promising for applications in extreme environments, involving high temperature, corrosion, and high ion-flux. In HECs, multiple principal cations form solid solutions, similar to medium/high-entropy alloys (M/HEA). However, mixing of atoms can be non-ideal, resulting in chemical short-range order (CSRO). CSRO has been already reported in M/HEAs, cation-disordered oxides, and high-entropy oxides and in many cases, it was found to have significant impact on materials properties. CSRO in covalently-bonded high-entropy ceramics has not been observed so far, and its potential impact on materials properties is unknown. In contrast to M/HEAs, in HECs only one of the sub-lattices forms a solid solution, and therefore it is unclear whether the concept of CSRO extends to HECs. Here, we report the observation of CSRO in multiple HECs using a combination of atomistic simulations and scanning transmission electron microscopy. We find that CSRO in HECs can be controlled by both selection of chemical elements and heat treatment, and it significantly improves radiation resistance, although it is not the only factor. Our findings expand the understanding of CSRO to HECs and provide a pathway for design of new materials for extreme environments.

HECs are an emerging class of materials with excellent properties, such as high hardness and ductility[1,2], superior radiation resistance[3–7], and high-temperature stability[2,8,9]. These properties are highly dependent on the ability to form a solid solution/single phase[10,11]. However, such mixing can be imperfect, leading to the potential formation of chemical short-range order (CSRO). Imperfect mixing has been recently discovered in other disordered systems, such as medium/high-entropy alloys (M/HEA)[12–22], Li-excess cation-disordered oxide[23,24], and high-entropy oxides[25]. In an earlier study[26], a distinction has been made between short-range ordering (SRO) and short-range clustering (SRC), which describes the attraction of unlike and like atoms, respectively. Here, we define the CSRO as the deviation of chemical order from a random arrangement on the same cation sublattice without altering

the lattice structure, which means that our definition includes both SRO and SRC. Due to the difficulty in characterizing CSRO experimentally[27,28], observations and understanding of CSRO in different systems are limited. For example, CSRO was investigated only via neutron diffraction[25] in high-entropy oxides (HEOs). However, HEOs exhibit a distinct bonding nature when compared to HECs. For instance, in HEOs, there is a significant electrostatic contribution to the interatomic bonding, which leads to preferential cation to oxygen bonding and may result in SRO. In contrast, in HECs, the bonding nature is mostly covalent, and therefore, it is not clear if the understanding of SRO in HEOs can be directly extended to HECs. Even in the more well-studied M/HEAs, there are contradictory reports regarding the ability to control CSRO. For example, Han Y. et al.[17] found that

[1]Department of Materials Science and Engineering, University of Wisconsin–Madison, Madison, WI, USA. [2]Department of Nuclear, Plasma & Radiological Engineering, University of Illinois Urbana-Champaign, Urbana, IL, USA. [3]School of Nuclear Science and Engineering, Shanghai Jiao Tong University, Shanghai, China. [4]Institute of Modern Physics, Fudan University, Shanghai, China. [5]Department of Nuclear Engineering and Engineering Physics, University of Wisconsin–Madison, Madison, WI, USA. [6]These authors contributed equally: Shuguang Wei, Muhammad Waqas Qureshi, Jingrui Wei, Longfei Liu. ✉e-mail: szlufarska@wisc.edu

varying chemical elements or thermal treatment history of M/HEAs had a negligible impact on CSRO, which disagrees with earlier work where substitutional elements[15,16] and thermal treatment[13] were used successfully to tune CSRO. HECs have more complex energy landscapes than metals and oxides due to a combination of metallic, covalent and ionic bonding. They also form multiple sublattices with distinct characters (e.g., anion vs cation), which leads to vastly different defect behavior on each of the sublattices. In addition, only one sub-lattice can form solid solutions in HECs, as the other sub-lattice consists entirely of carbon in the absence of impurities. The formation of CSRO in such systems, the ability to tune it, and the impact of CSRO on properties are all open questions.

To answer these questions, we study two HEC compositions: (TiVZrNb)C and (TiVMoNb)C, labeled HEC-Zr and HEC-Mo, respectively. Our atomistic simulations not only reveal the existence of CSRO in both compositions, but also predict two methods to tune the CSRO, namely the elemental selection and the thermal treatment. CSRO in experimental samples is confirmed using a combination of atomic resolution high-angle annular dark-field (HAADF) imaging as well as 4-dimensional scanning transmission electron microscopy (4D-STEM). A significant reduction of CSRO with the increase of annealing temperature is observed in HEC-Mo, both in simulation and experiment, though the temperature regime is not the same in the two approaches. To reach the low CSRO state, HEC-Mo is annealed at 1773 K both in simulation and experiment. The high CSRO state in simulated HEC-Mo is created at 300 K to make the CSRO most pronounced. However, 300 K is not practical for experimental studies of HEC-Mo due to the low diffusivity of atoms. Thus, HEC-Mo sample is annealed at 1573 K, where CSRO exists and kinetics of CSRO are high enough to form it, and compared to HEC-Mo annealed at 1773 K, where sample prefers a more random state. The reduction of CSRO is also observed, in qualitative agreement with simulations. With the combination of simulations and experiments, we explore the existence of CSRO in HEC, two ways to tune it, and its impact on material properties. The understanding of CSRO paves the way for designing high-performance HECs.

## Results

### Observation of CSRO in HECs

To simulate the HECs, we first developed a machine learning interatomic potential (MLIP) based on the moment tensor potential (MTP) formalism. Subsequently, hybrid molecular dynamics/Monte Carlo (MD/MC)[17,29] simulations were employed to minimize energy and find the corresponding atomic arrangements in HEC-Zr and HEC-Mo. After 6 million MC swaps, all simulations have reached equilibrium states (Fig. S1). Simulations predict that CSRO forms in these materials, but to varying degrees. Atomic arrangements and Warren-Cowley (WC) parameters[30,31] up to 3rd NN in cation sublattice for the simulated HECs are shown in Fig. 1 and Fig. S2. The WC parameter $\alpha_{ij}$ represents the propensity for atoms $i$ and $j$ to attract each other in the solution, with $\alpha_{ij} = 0$ representing random arrangement, $\alpha_{ij} < 0$ attraction, and $\alpha_{ij} > 0$ repulsion of the elements. As illustrated by the WC parameter, the chemical preference of atomic pairs is most obvious among the 1st (Fig. 1b, d, f) and 2nd NN (Fig. S2). The ordering is almost negligible for the 3rd NN atoms (Fig. S2), indicating no long-range order formation. Since the 1st NN dominates the local environment and thus impacts material properties, in the following analysis we focused on the 1st NN.

In HEC-Zr, the chemical preference of V-V pairs is the strongest ($\alpha_{ij} = -0.53$) (see Fig. 1b). In HEC-Mo, the V-V pair still exhibits the strongest chemical preference among all the elements, but the attraction is weaker than in HEC-Zr, with $\alpha_{ij} = -0.26$ (Fig. 1d). In HEC-Zr, the Zr-Zr pair also exhibits strong chemical preference, but when Zr is replaced with Mo, such preference disappears as Mo-Mo pairs are repulsive ($\alpha_{ij} = 0.37$). Apart from SRC, SRO also exists and is comparable between HEC-Zr and HEC-Mo. For instance, in HEC-Zr, Nb-Ti ($\alpha_{ij} = -0.28$) and Nb-Zr ($\alpha_{ij} = -0.17$) pairs exhibit strong attraction. In

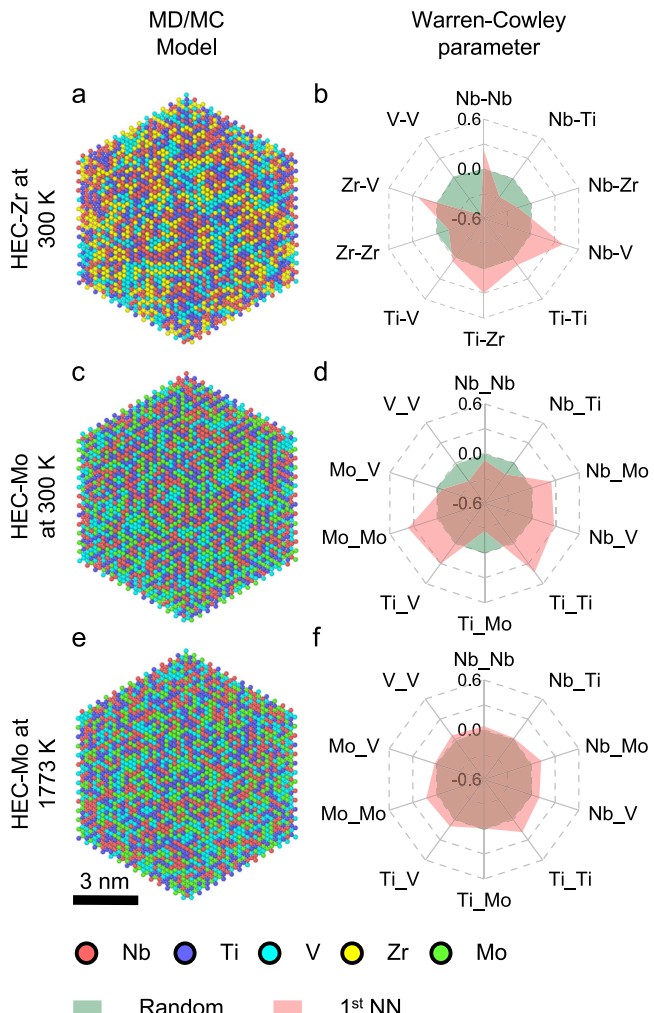

MD/MC Model

Warren-Cowley parameter

HEC-Zr at 300 K

HEC-Mo at 300 K

HEC-Mo at 1773 K

3 nm

Nb ⬤ Ti ⬤ V ⬤ Zr ⬤ Mo ⬤

Random 1st NN

**Fig. 1 | Visualization of CSRO in simulated HEC systems.** Visualization of atomic arrangements in (**a**) HEC-Zr at 300 K, (**c**) HEC-Mo at 300 K, and (**e**) HEC-Mo at 1773 K. The Warren-Cowley parameter when considering the 1st nearest neighbor (NN) metal-metal interaction within the cation sublattice of (**b**) HEC-Zr at 300 K, (**d**) HEC-Mo at 300 K, and (**f**) HEC-Mo at 1773 K. The Warren-Cowley parameter represents the chemical preference for two atoms and it is defined in the main text. Source data are provided as a Source Data file.

HEC-Mo, strong chemical preference can be observed for Nb-Ti ($\alpha_{ij} = -0.17$) and Ti-Mo ($\alpha_{ij} = -0.26$). Overall, the degree of CSRO in HEC-Zr is stronger than that in HEC-Mo, where the degree of CSRO is measured by the deviation of chemical order from a random distribution. In addition, our simulations predict that heat treatment will reduce the degree of CSRO as evidenced by simulated annealing at two different temperatures (Fig. 1c–f). With the increase of annealing temperature, the CSRO is significantly reduced and almost disappears, as evidenced by the near-zero WC parameter. A more complete temperature dependence of CSRO in simulated HEC-Mo is plotted in Fig. S3a. It shows that with an increase of temperature, CSRO gradually decays and significantly diminishes after 900 K.

To validate our prediction on the existence of CSRO and pathways to tune them in HEC from MD/MC simulations, HECs were heated in a differential thermal analyzer (DTA) to monitor the energy change. More specifically, the formation of CSRO leads to heat release, resulting in an exothermic peak, and the dissolution of CSRO requires heat absorption, giving rise to an endothermic peak. In HEC-Zr, an exothermic peak starts at ~1473 K, indicating the initiation of CSRO formation (Fig. 2a). This observation suggests that below 1473 K, the kinetics for CSRO in HEC-Zr are frozen (Fig. S3b). At ~1563 K, an

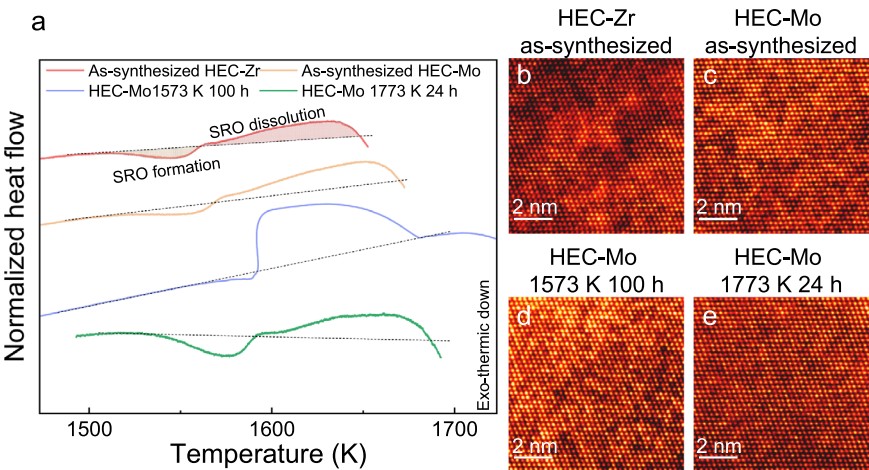

**Fig. 2 | Evidence of CSRO in HECs. a** Normalized heat flow vs. temperature measured in DTA during heating, showing the formation of CSRO and the dissolution of CSRO. Shaded areas represent the amount of heat released and absorbed during formation and dissolution of SRO, respectively, and are shown for as-synthesized HEC-Zr, as an example. **b**–**e** Z-contrast high-angle annular dark-field images of **b** as-synthesized HEC-Zr with strong intensity variation, **c** as-synthesized HEC-Mo with weaker variation, **d** HEC-Mo annealed at 1573 K for 100 h with comparable variation to HEC-Mo shown in (**c**), and **e** HEC-Mo annealed at 1773 K for 24 h with almost uniform contrast. Source data are provided as a Source Data file.

endothermic peak emerges, indicating that energy change is dominated by the dissolution of CSRO. The endothermic peak ends at ~1653 K, which suggests that at this temperature, the extent of CSRO in HEC-Zr is significantly decreased. The fact that the endothermic peak is larger than the exothermic peak in HEC-Zr suggests there is pre-existing CSRO, which could be explained as follows. The as-synthesized samples are quenched from 2573 K, where a partial melt may form (Fig. S11c), and these samples may not reach fully homogenous states. The CSRO observed in as-synthesized samples is likely to originate from the solidification process, as pointed out by Han et al.[17]. Interestingly, Han et al.[17] found that once CSRO formed in solid solution, the chemical preference would be similar, regardless of whether CSRO originated from quenching or annealing. In as-synthesized HEC-Mo, both exothermic peak and endothermic peak can be observed, with the latter having a larger area. Thus, one can conclude that there is pre-existing CSRO in as-synthesized HEC-Mo as well. It is worth noting that there is a discrepancy between experimental and simulation temperatures above which CSRO is largely dissolved, which is not necessarily surprising for an interatomic potential (Fig. S3). However, it is not a concern for analysis as long as the simulated systems are chosen below and above the temperature where CSRO drops significantly according to the interatomic potential. To further validate our DTA method, we calculated the enthalpy changes in HEC-Zr (Fig. S4a) and HEC-Mo. These changes were induced by the formation and dissolution of CSRO and were calculated based on the exothermic and endothermic peak areas, which equal ~56.5 meV/atom and ~62.0 meV/atom, respectively. These values agree well with enthalpies calculated using DFT and methods from our previous work[32] (60.78 meV/atom for HEC-Zr and 48.93 meV/atom for HEC-Mo), providing further support for the claim that the observed peaks in DTA curves are induced by the formation and dissolution of CSRO. The positive values of formation enthalpies also demonstrate that formation enthalpy is the driving force for the formation of CSRO.

We have also used DTA to explore the temperature dependence of CSRO in HEC-Mo, which has been predicted by our simulations. Toward this end, HEC-Mo was first annealed at two different temperatures to induce different levels of CSRO (Fig. S3b) and then quenched to room temperature. The degrees of CSRO in HEC-Mo were then determined by DTA. For example, for HEC-Mo annealed at 1573 K, where a higher level of CSRO is preferred (Fig. S3b), DTA curve exhibits a minimal CSRO formation (exothermic) peak followed by a significantly larger CSRO dissolution (endothermic) peak (Fig. 2a), suggesting there is a great amount of pre-existing CSRO, which formed during annealing. In contrast, for HEC-Mo annealed at 1773 K, where CSRO is expected to be significantly reduced (Fig. S3b), DTA curve exhibits a large CSRO formation (exothermic) peak (Fig. 2a), followed by a CSRO dissolution (endothermic) peak that is only slightly larger than the formation (exothermic) peak (Fig. 2a), suggesting that the amount of pre-existing CSRO is relatively low. In other words, HEC-Mo annealed at 1773 K exhibits a considerably reduced amount of CSRO when compared to HEC-Mo annealed at 1573 K.

After the validation of the existence of CSRO in HECs, atomic resolution HAADF was used on HECs to visualize CSRO. Z-contrast images acquired by HAADF have been successfully used to characterize composition fluctuations in other materials[23]. This analysis applied to HECs (Fig. 2) shows a clear contrast variation (Fig. 2), though the extent of the variation and the domain sizes vary among the samples. In HEC-Zr, the contrast variance is the strongest, with the sizes of bright and dark domains falling into the range of 1-2 nm (see Fig. 2b). This observation suggests there exists local clustering of heavy and light atoms, which is short-range ordering. Comparison to MD/MC simulations with CSRO suggests that the bright regions are Zr-rich and dark regions are V-rich. Meanwhile, the strongest variation in contrast is a result of the strongest clustering preference of Zr-Zr and V-V pairs. In HEC-Mo, a contrast variation is still present (see Fig. 2c), though weaker and smaller than in HEC-Zr. Compared to our simulations with CSRO, the dark region in HEC-Mo is still V-rich, while the bright region is Nb-rich. This analysis agrees with predictions from our simulations (Fig. 1) that the degree of CSRO is weaker in HEC-Mo than in HEC-Zr, and provides evidence that elemental selection of metal cations can be used for tuning CSRO. The significant reduction of CSRO with the increase of temperature observed in simulations and DTA is further visualized by STEM. More specifically, after annealing at 1573 K, Z-contrast imaging show contrast variance (Fig. 2d), which is comparable to as-synthesized HEC-Mo (Fig. 2c). However, after annealing at 1773 K, the contrast variation in HEC-Mo almost disappeared (see Fig. 2e), indicating a significant drop in CSRO. It is important to mention that we excluded the possibility that the Z-contrast shown in Fig. 2 arises from variations in sample thickness, surface oxidation, or strain. Specifically, we ensured consistent thickness in the acquisition areas, removed potential surface oxides though ion milling before the acquisition, and observed consistent image

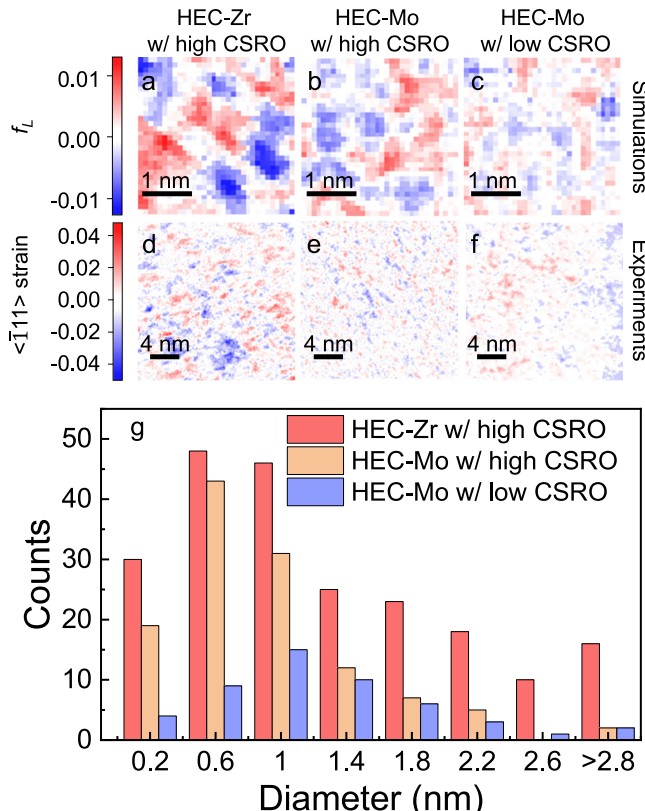

**Fig. 3 | Strain analysis of both simulated and experimental HECs. a–c** Spatial distribution of fractional change in the average metal-metal bond length ($f_L$) in simulated HEC-Zr with high CSRO, HEC-Mo with high CSRO, and HEC-Mo with low CSRO, respectively, at [011] zone axis. $f_L$ represents the local strain and it is defined in the main text and in the method section. **d–f** Spatial distribution of strains along [$\bar{1}$11] direction in experimental HEC-Zr with high CSRO (as-synthesized), HEC-Mo with high CSRO (as-synthesized), and HEC-Mo with low CSRO (annealed at 1773 K), respectively, at [011] zone axis. Strains are calculated from diffraction patterns collected using 4D-STEM with cepstrum analysis. The [$\bar{1}$11] direction was chosen as this is the direction of the nearest metal-metal bonding, where the strain is directly related to the clustering of cations. **g** Histograms of identified strain domain diameters in experimental HECs. The strain domains were defined as region with strain with absolute value larger than 1.5 %. Complete numerical data contributing to histogram are provided in the Source Data. Source data are provided as a Source Data file.

contrast over large areas acquired at different collection angles to exclude the effect of strain. The averaging effect of sample thickness was also considered but due to the electron channeling[33,34], it does not impact the measurement of CSRO (See Supplementary note 1).

As the Z-contrast images can only qualitatively reveal CSRO, we next aimed to characterize CSRO more quantitatively, targeting both compositions with high CSRO (as-synthesized) and HEC-Mo with low CSRO (annealed at 1773 K). Toward this end, we employed 4D-STEM to collect nanobeam electron diffractions (NBED) along the [011] zone axis, which were then processed with cepstrum analysis for more precise lattice structure measurement[35,36]. Probing CSRO based on 4D-STEM is challenging because it is not obvious which features in the analysis show the strongest correlation with CSRO. To answer this question, we first analyzed our simulated HECs, with a particular focus on the lattice strain as it has been shown in earlier study to correlate with CSRO in M/HEA[18,37]. Interestingly, we found that in simulated HECs, local lattice strain is distributed heterogeneously with domain size in the range of 1-2 nm, which is the same size as the domain observed in Z-contrast images. More specifically, the fractional change of the average bond length ($f_L$) between the nearest metal-metal pairs

was calculated and plotted in Fig. 3a–c. Here, $f_L$ represents the deviation of the average metal-metal bonding length in a local region from the average bonding-length in the simulation supercell. This quantity can be regarded as a local lattice strain, with $f_L = 0$ representing no local strain, $f_L > 0$ representing local tension, and $f_L < 0$ representing local compression. In HEC-Zr with high CSRO (see Fig. 3a), the strain variation is the strongest, suggesting the strongest tendency for atoms to cluster and form CSRO. Detailed analysis (Fig. S6a) indicates that the clustering of Zr induces tensile strain and the clustering of V results in compression. In HEC-Mo with high CSRO (see Fig. 3b), peak strains decreased, which confirms weakening of CSRO in HEC-Mo relative to HEC-Zr. Detailed analysis (Fig. S6b) reveals that clustering of V still accounts for compression, whereas the clustering of Nb induces tensile strain. After simulated annealing at 1773 K, lattice strain dropped dramatically in HEC-Mo, consistent with significant reductions in CSRO. In essence, heterogeneous strain in HECs is tightly correlated with CSRO and can be employed for the detection of CSRO.

Strains along the [$\bar{1}$11] direction were then calculated for experimental HECs using [011] NBED patterns with cepstrum analysis to probe CSRO. [$\bar{1}$11] direction was picked as it intersects with all directions of the nearest metal-metal bonding. Spatial distributions of strains are illustrated in Fig. 3d–f. Discernible heterogeneous strain can be observed in all three types of HECs, but to varying degrees, as reflected in the different sizes and numbers of strain domains. For example, in HEC-Zr with high CSRO, the maximum strain is the largest, with +5.2% for tensile strain and −5.7% for compression (see Fig. 3d). In HEC-Mo with high CSRO, maximum strain drops to +3.8% for tension and −4.6% for compression (see Fig. 3e). After annealing at 1773 K, maximum strain in HEC-Mo further decreased to +2.6% for tension and −2.3% for compression, indicating a significant reduction of CSRO. Heterogeneous strain in experimental HECs agrees well with the degree of CSRO in simulated HECs, further corroborating that the heterogeneous strain provides a reliable probe of CSRO in HECs. In addition, 4D-STEM analysis of CSRO shows good agreement with the observation from HAADF, providing further evidence that the differences in contrast variance of Z-contrast images are induced by CSRO.

We note that the domains of heterogeneous strain in experimental HECs are generally larger than those in simulated HECs, which is likely due to several factors. Here, we speculate on two specific reasons. First, electron probe will convolve and blur the features measured experimentally. Second, the strain induced by CSRO can extend beyond the region of CSRO, further enlarging the domain size. To determine the domain size of CSRO, we selected a threshold value of strain to be 1.5 % in absolute value. This value is slightly higher than the lattice strain in simulated fully random HECs. The domain diameters were then calculated by assuming spherical shapes. Experimental size distributions of CSRO domains are shown in Fig. 3g. Not surprisingly, HEC-Zr with high CSRO exhibits the largest average domain size of 1.4 nm and the highest domain density of $4.6 \times 10^{17}\,\text{m}^{-2}$. In HEC-Mo with high CSRO, the average domain size decreases to 0.9 nm and the number density dropped to $2.5 \times 10^{17}\,\text{m}^{-2}$, implying a weakened CSRO. After annealing at 1773 K, the average domain size in the HEC-Mo changed to 1.2 nm, but it is clear from Fig. 3g that the overall CSRO decreased, as quantified by the number density which dropped to $1.1 \times 10^{17}\,\text{m}^{-2}$. The domain sizes are consistent with the distance to the second or the third atomic shell of the metal sublattice, providing further evidence that the ordering is short-range. It is worth mentioning that, during the collection of 4D-STEM, no second phase was observed in HECs, further proving the formation of CSRO does not alter the lattice structure. For that reason, the CSRO in HECs is distinct from the SRO in complex oxides, like isometric pyrochlore[38,39], where SRO involves a phase transformation in the range of one to two unit cells.

To summarize, we have predicted using simulations that CSRO is ubiquitous in HECs, with elemental selection and thermal treatment as

pathways to tune the degree of CSRO. The combination of HAADF imaging and strain analysis based on 4D-STEM confirms predictions from our models.

## CSRO-induced improved radiation resistance

Having demonstrated the existence of CSRO in HECs, we ask whether the presence of CSRO is relevant for physical properties of the materials. We have specifically focused on resistance to radiation since in M/HEAs, CSRO has been proposed to improve radiation resistance by impeding the diffusion of defects[40,41]. Toward this end, we have irradiated HEC-Zr and HEC-Mo with high CSRO (as-synthesized) and HEC-Mo with low CSRO (annealed at 1773 K) with 4.5 MeV Si ions and quantified the resulting radiation damage using radiation-induced swelling determined from the lattice expansion measured via grazing incidence X-ray diffraction (GIXRD). The results are summarized in Fig. 4a, which also includes swelling values reported in earlier studies on HEC[7,42–46]. The lattice expansion of our samples falls within the range of 0.15 %-1.2 %, which is comparable to previous work. For all samples, increasing temperature leads to a decrease in lattice expansion, which can be attributed to dynamic annealing of defects[46]. The order of the samples with respect to swelling follows the same qualitative trend at each temperature, so we focus on irradiation at 573 K as a representative example. As shown in Fig. 4a, the order of samples from the highest to the lowest lattice expansion is HEC-Zr with high CSRO (1.16%), HEC-Mo with low CSRO (0.84%), and HEC-Mo with high CSRO (0.38%). We have checked the grain size in HEC-Mo with high CSRO and HEC-Mo with low CSRO using electron backscatter diffraction (Fig. S7) as grain boundaries can serve as defects sink and impact defect recovery. The grain size is very similar in the HEC-Mo samples with different levels of CSRO, which means that we can exclude the possibility that the observed variation in radiation resistance is due to the grain size. Comparing the two HEC-Mo samples allows us to isolate the effect of CSRO as the overall chemistry of the samples is the same. We can therefore confidently conclude that CSRO led to significant improvement in radiation resistance in HEC. The stability of CSRO under radiation remains an open question and it will likely depend on the radiation flux and temperature.

Interestingly, HEC-Zr with high CSRO has the highest degree and the largest number density of CSRO, but its resistance to radiation-induced swelling is the worst among the HECs studied in this work. This result suggests that CSRO is not the only factor impacting radiation resistance in HECs but that chemistry also plays an important role as well. Here, by chemistry, we mean the stoichiometry on the cation sublattice. The effect of chemistry in the anion sublattice (carbon stoichiometry) on radiation resistance in HECs is negligible. Specifically, our XPS measurements indicate that both HEC-Zr and HEC-Mo are C-rich, with carbon concentration equal 56.72 at.% in HEC-Zr and 52.60 at.% in HEC-Mo (Table S1). Previous study[47] does suggest that graphite can appear in C-rich ZrC, which serves as a strong source of knock-on carbon interstitials into carbides and lead to larger dislocation loops. However, our simulation suggested that C interstitials in HECs share comparable migration energies (3.56 eV for HEC-Zr and 3.86 eV for HEC-Mo) and are immobile at the irradiation temperature of the current study. Thus, if graphite were present in these samples and introduced some C interstitials into HECs, these interstitials could not migrate and coalesce to larger dislocation loops and the impact on lattice expansion would be minimal. A detailed study of the mechanisms responsible for the impact of CSRO on radiation resistance is beyond the scope of this paper. However, TEM analysis of irradiated HEC-Mo samples (Fig. 4b, c) reveals that defects in HEC-Mo with high CSRO are qualitatively smaller than the defects in HEC-Mo with low CSRO. In addition, small dislocation loops were observed in HEC-Mo with low CSRO (Fig. 4c) whereas only small defect clusters (black-spot defects) appear in HEC-Mo with high CSRO (Fig. 4b), suggesting coarsening of defects in HEC-Mo with low CSRO. Based on the TEM

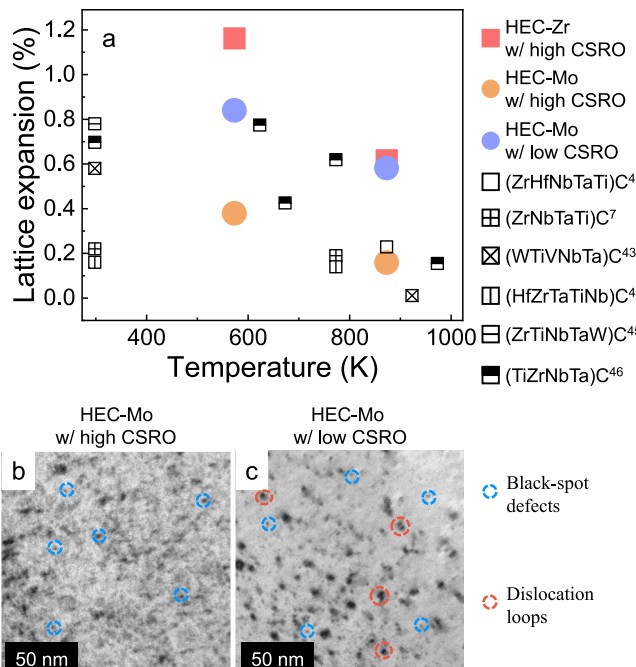

**Fig. 4 | Radiation-induced swelling in HEC-Mo. a** Lattice expansion of carbides measured using GIXRD. Full symbols correspond to data from this study, while open black symbols are data collected from literature[7,42–46]. For our samples, squares represent HEC-Zr and circles represent HEC-Mo. Bright-field TEM images of HEC-Mo with high CSRO **b** and HEC-Mo with low CSRO **c** under two-beam conditions with a g vector of <11$\bar{1}$> near the [112] zone axis. Source data are provided as a Source Data file.

observations, the distinct defect behavior is likely due to the different defect kinetics induced by CSRO.

In summary, we have discovered CSRO in HECs and demonstrated that it can significantly improve the resistance to radiation-induced damage in HECs, although it is not the only factor. CSRO was also identified to be ubiquitous in cation sublattices of HECs, similar to M/HEA[17], even though HECs have more complex energy landscapes. Our finding suggests that CSRO may not be limited only to certain systems[12–22], but instead it may be universal in all high-entropy materials. Further studies will be needed to determine the mechanisms underlying the impact of CSRO on properties of HECs and similarities and differences between HECs and M/HEA with respect to these mechanisms.

## Methods

### Fitting machine-learning interatomic potential (MLIP)

The MLIP model, based on moment tensor potential (MTP) formalism was employed, as it offers a good tradeoff between accuracy and computational efficiency[48]. Details on the MTP formalism are available in refs. [49–51]. The training data for HEC-Zr and HEC-Mo were generated from elemental systems along with unary, binary, ternary, and quaternary carbides. To generate the training data, we ran MD simulations with M3GNET[52] universal potential using LAMMPS software[53]. The M3GNET universal potential, while useful for running MD simulations for the purpose of generating atomic configurations, may not be accurate enough for the purpose of our simulations; therefore, we trained a potential specifically to model our systems. For each crystalline phase we performed a simulation in the temperature range of 0 K to 5000 K and pressure range of −1 GPa to 1 GPa for 100 ps. Then, a total of 10 to 15 atomic configurations were extracted at equal time spans during the simulation and we calculated their energies and atomic forces using density functional theory (DFT) to generate a

complete training data. The detailed structure generation for HEC-Zr is as follows. A similar strategy was used for HEC-Mo. Systems included in our training data are

### 1. Elemental and unary carbide system.
  (a) Ground state structure for the elements i.e., BCC Nb, Mo V, and HCP Ti, Zr, C and unary carbides NbC, TiC, ZrC, MoC, and VC.
  (b) Snapshots from MD simulations for each system.

### 2. Binary and ternary carbide systems.
  (a) Special quasi-random structure (SQS) generated solid solution of all binary combinations ((NbTi)C, (NbZr)C, (NbV)C, (TiZr)C, (TiV)C, (ZrV)C), and ternary combinations ((NbTiZr)C, (NbTiV)C, (NbZrV)C, (TiZrV)C). To limit the compositional space, the number of atoms was kept to 64 (32 metal atoms, 32 C) in each cell. The composition of each carbide was generated by partial substitution of one element with the other in the form of $(A_xB_{1-x})$C with $x$ in the range 0-100 at% at interval of 6.25 at %.
  (b) For each binary and ternary system, snapshots from MD simulations were added. Three different solid solutions were selected: the end members and one composition in the middle.

### 3. Quaternary carbide systems.
  (a) Ground state SQS permutations of quaternary (NbTiVZr)C and (NbTiVMo)C carbides were considered.
  (b) Random compositions of quaternary carbides were selected from MD simulation and snapshots and added to training data.

All the abovementioned atomic configurations were included in DFT calculations of energies and forces. All the DFT calculations were performed using the Vienna Ab-initio Simulation Package (VASP) software[54]. The same convergence criteria were set to get energy and forces for all the atomic configurations. The projected-augmented-wave with the Perdew-Burke-Ernzerhof (PBE) form of exchange-correlation potential was used in this study. The energy cutoff of 500 eV was used for the plane-wave basis set with Monkhost-Pack k-point mesh of $3 \times 3 \times 3$.

## Molecular dynamic simulations
All the atomistic simulations were performed using our MTP potential. A $14 \times 14 \times 14$ rock-salt supercell was initially constructed, with metal atoms randomly distributed in equiatomic compositions (random state). To equilibrate the CSRO, hybrid MD/MC simulations were performed with periodic boundary conditions in all directions. The systems were first relaxed and equilibrated to 300K for both HEC-Zr and HEC-Mo, and multiple temperatures (600K, 900K, 1200K, 1573K, and 1773K) for HEC-Mo and zero pressure in the canonical ensemble. Subsequently, MC swaps were conducted in which the position of two randomly selected atoms of different types were accepted and rejected based on the Metropolis algorithm. To relax the atomic configurations, MD simulations were performed in between the MC swaps, with 100 MD steps for every 100 MC swaps. The relative mole fractions of different atom types were kept constant. A total of 6 million MC swaps were performed for each HEC system until the equilibrium state was approached (Fig. S1). The final atomic configuration was then considered for CSRO analysis.

To quantify the CSRO, we calculated Warren-Cowley parameter[30,31] (WCP) for each pair of atoms i and j, represented as $\alpha_{ij}$:

$$\alpha_{ij} = 1 - \frac{\langle N_{ij} \rangle}{NX_j} \qquad (1)$$

where $\langle N_{ij} \rangle$ is the average number of type j atoms surrounding type i atoms within the 1st NN within cation sublattice, N is the coordination number in 1NN, and $X_j$ represents the concentration of atom j in the given HEC system.

## Sample preparation and ion irradiation
For the study of CSRO and its impact on resistance to radiation-induced damage, we targeted (TiVZrNb)C and (TiVMoNb)C for the following reasons. First, we targeted refractory metals as the corresponding binary carbides exhibit ultra-high melting temperatures, making them promising candidates for high-temperature applications. Second, Zr was selected because ZrC has been widely studied for potential applications in nuclear reactors. Among many ZrC-based HECs, we selected (TiVZrNb)C as it has been already synthesized in earlier studies[55] and because TiC, VC and NbC have also been considered as potential candidates in nuclear industries. Third, to explore the effect of elemental selection on CSRO, Zr was replaced by Mo and (TiVMoNb)C was chosen. After selecting (TiVZrNb)C and (TiVMoNb)C, five binary carbide powders (TiC, VC, ZrC, NbC, $Mo_2C$) and graphite powders with purity larger than 99 at.% were used as precursors for the synthesis of high entropy carbides. The carbides powders were weighed in an equimolar way for the cations to achieve the desired compositions for a total mass of 15 g 0.5 vol.% graphite powder was also added to compensate for the carbon deficiency as well to remove surface oxycarbide layers. Powder mixtures were then ball milled in $Si_3N_4$ jar with $Si_3N_4$ balls in an ethanol medium. All ball millings were performed in a sequence of 5 minutes of milling and 5 minutes of rest to avoid overheating for a total of 24 hours. Mixed powders were dried at 338 K and then sieved with a 100 mesh. Two-step spark plasma sintering was then employed to facilitate the conversion of carbide into single-phase carbide and densify powders to a solid compact, simultaneously. Entire sintering process is performed under backfilled argon to avoid oxidation. In the first step, powders were heated to 2273 K with heating rate 100 K·min⁻¹ and dwell at 2273 K for 10 minutes. Simultaneously, pressure applied on powders was increased to 70 MPa at a rate of 10 MPa·min⁻¹. In the second step, the temperature was further increased to 2573 K at the same ramping rate and dwell time of 2 minutes at this peak temperature. Then, the power of the furnace was turned off, and the furnace was cooled by flowing water from a chiller. All sintering experiments were performed in 20 mm graphite die and punch sets with graphite foil for the separation between powders and mold. Parts of the (TiVMoNb)C sample synthesized using SPS was further annealed separately at 1573 K for 100 h and at 1773 K for 24 h followed by furnace cooling. The reasons for choosing different thermal conditions to study the temperature dependence of CSRO in simulations and experiments are the limitations of CSRO kinetics in experiments and the discrepancy between experimental and simulation temperatures above which CSRO is largely dissolved. All samples are cut into 5 mm × 2 mm × 2 mm cuboids using a wafering blade and mirror polished on one side using diamond lapping film discs.

Polished samples were further irradiated with 4.50 MeV Si ions to a total fluence of $1.96 \times 10^{16}$ ions·cm⁻² at two different temperatures, 573 K and 873 K using the tandem accelerator at Ion Beam Laboratory at the University of Michigan. The ion flux for irradiation experiments was set at $1.09 \times 10^{12}$ ions·cm⁻²s⁻¹, corresponding to a duration of 5 hours. SRIM-2013[56] was employed to simulate the radiation damage, measured in dpa, in all carbides, with threshold energies for all atoms set at 25 eV Based on SRIM calculations, the damage level reaches its peak of ~12 dpa at the depth of ~1.6 μm below the surface (Fig. S8a). At the first 600 nm below the surface, damage level as a function of depth is relatively flat (damage plateau) and the average damage level in this region is ~1.5 dpa. We can exclude the potential

chemical effect of Si by calculating implantation profile using SRIM-2013[56]. For example, as simulated by SRIM (Fig. S8b), the highest concentration of Si in HECs is less than 0.8 at.%, suggesting minimal chemical effects induced by Si.

### Differential thermal analysis (DTA)

Perkin Elmer DTA 7 differential thermal analyzer was employed to determine the formation and dissolve of SRO in HECs. HECs with weight of 10-30 mg were heated from 300 K to 1773 K and held at peak temperature for 10 minutes then cooled down from 1773 K to 300 K. The cycle of heating and cooling were repeated 3 times to validate the reproducibility of the measurements (Fig. S4b). The heating rate and cooling rate were kept at 20 K/min. All experiments are performed under a flow of Ar gas with a flow rate of 50 mL/min. Measured heat flow is normalized by sample weight.

### High-resolution STEM (HRSTEM) imaging and 4D-STEM characterization

We prepared electron-transparent samples for all TEM and STEM characterizations performed in this work using a standard lift-out technique using an FEI Helios PFIB G4 FIB/FESEM Focused Ion beam (FIB) in the Nanoscale Imaging and Analysis Center, University of Wisconsin-Madison, which has been described in detail in our previous work[57]. Further thinning was performed using a Fischione 1040 TEM Ion Mill with an 800 V ion beam. The inelastic mean free space for HEC-Zr and HEC-Mo were estimated to be 82.9 and 82.6 nm based on the model proposed by Malis et al.[58] and Egerton et al.[59]. TEM specimen thickness measured using EELS log-ratio method are estimated to be 41.4 nm, 39.6 nm, 29.7 nm and 37.2 nm for HEC-Zr, HEC-Mo, HEC-Mo annealed at 1573 K, and HEC-Mo annealed at 1773 K, respectively. All atomic resolution HAADF images and 4D STEM analysis were collected using an FEI Titan G2 80-200 TEM (operated at 200 kV) equipped with a CEOS probe corrector. High-angle annular dark-field (HAADF) image series were acquired with a 23.4 mrad semi-convergence angle and a 147 pA current probe. Nonrigid registration[60] was used to compensate for the drift and the series was averaged to get a single frame with a high signal-to-noise ratio. The image exposure time and thickness of study regions for different specimens were kept consistent to achieve meaningful comparison. For 4D-STEM imaging, we use a Direct Electron Celeritas-XS camera to collect convergence beam electron diffraction (CBED) pattern with a convergence angle of 3 mrad or 1.5 mrad and a camera length of 60 mm. The camera was operated at 6000 frames per second, and 10 4D STEM scan series were collected and registered for a better signal-to-noise ratio. The thicknesses of the regions studied by 4D-STEM were calculated based on the zero-beam vs {111} diffraction beam intensity ratio, which depends on thickness due to dynamical scattering[61]. As illustrated in Fig. S9, the measured thickness is 30.4 nm, 36.0 nm, and 43.6 nm for HEC-Zr, HEC-Mo, and HEC-Mo annealed at 1773 K, respectively. These values are reasonably close to the EELS estimates given above. The CBED patterns were then analyzed using the cepstrum method developed in previous work[35]. Details of the cepstrum analysis implementation are included in Supplementary Note 2 and Fig. S10. To calculate fractional change ($f_L$) in nearest metal-metal interaction distance (1$^{st}$ NN of cation sublattice), simulated HECs were divided into a series of cylindrical sub-volumes ($v_i$) which approximates the region interacted with the STEM electron beam, each with a diameter of 6 Å, oriented along [011], positioned on a fine sampling grid with 1 Å spacing. The average bonding length of cations ($\langle L_{MM} \rangle_{v_i}$) in each sub-volume was first calculated. Then the average bonding length of cations ($\langle L_{MM} \rangle_V$) over the entire supercell. Finally, the fractional change in nearest metal-metal interaction distance (1$^{st}$ NN of cation

sublattice) $f_L$ can be calculated by the following equation:

$$f_L = \frac{\langle L_{MM} \rangle_{v_i} - \langle L_{MM} \rangle_V}{\langle L_{MM} \rangle_V} \qquad (2)$$

### Characterization of HECs before and after irradiation

XRD experiments were performed using a Bruker D8 discovery X-ray diffractometer equipped with a two-dimensional detector to confirm the synthesized materials are single-phase. For all acquisitions, the scan range (2θ) was set to be 20°–80° with a step of 20° and at least 120 s for each step Fig. S11). To confirm that cations are uniformly distributed in HECs, EDS area scans were performed on HECs with an FEI Helios PFIB G4 equipped with the Thermo Noran Energy dispersive X-ray microanalysis system, operated at 15 kV. Based on EDS mapping (Fig. S11), no elemental segregation or precipitation was observed in HECs. Compositions were further measured using X-ray Photoelectron Spectroscopy (XPS). The measured region was first cleaned using Ar sputter cleaning to remove any carbon contamination and high-resolution spectra of C 1$s$, Ti 2$p$, V 2$p$, Zr 3$d$, Nb 3$d$ and Mo 3$d$ regions were collected from the cleaned region. Spectra were averaged over 15 scans with a pass energy of 50 eV and a step size of 0.2 eV. The composition of each element was calculated and included in Table S1. From our XPS measurements, both samples are enriched in carbon, but we did not observe any graphite formation. If such graphite formed, its impact on the conclusions presented in this paper would be minimal, as explained earlier.

To evaluate the radiation induced damage in the near surface region, GIXRD was performed on each sample before and after irradiation. The scan range was set to be 25°–65° at an incident angle of 3° with a step size of 20°. For each step, a long acquisition time of 600 s was employed to ensure a high signal-to-noise ratio. Collected data Fig. S12) were further fitted using Rietveld analysis to obtain the lattice parameter.

Detailed characterizations of radiation-induced defects were achieved using bright-field imaging collected at two-beam conditions with a FEI Tecnai TF 30 TEM operated at 300kV. We focused on the peak damage region for a better characterization of radiation-induced defects. Two-beam condition images were collected with a g vector of <11$\bar{1}$> near the [112] zone axis.

## Data availability

Source data are provided with this paper. All data that support this study are presented in the main text and/or the Supplementary Information and are available from the corresponding author upon request. Source data are provided with this paper.

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

## Acknowledgements

I.S., S.W., and M.W.Q. gratefully acknowledge support from the Department of Energy Basic Energy Science Program (grant # DEFG02–08ER46493). J.W. and P.M.V. acknowledge support for STEM experiments and simulations from the Harvey D. Spangler Professorship at UW-Madison and from the National Science Foundation (OAC-1931298) for preparation of the STEM datasets for dissemination. S.W. acknowledges the help from Fengdan Pan for training on ball milling. This work used the TACC's Stampede3 at the University of Texas at Austin through allocation TG-MAT240078, from the Advanced Cyber-infrastructure Coordination Ecosystem: Services & Support (ACCESS) program[62], which is supported by National Science Foundation (NSF) grants #2138259, #2138286, #2138307, #2137603, and #2138296.

## Author contributions

S.W., M.W.Q., and I.S. conceived the project, and I.S. supervised the project. J.W. and S.W. designed and performed the 4D-STEM and J.W. performed 4D-STEM analysis work. M.W.Q., J.X., and S.A. trained the MLIP, and M.W.Q. conducted DFT and MD simulations. S.W., X.H., L.L., E.W., R.S., H.Z., X.W., K.S., and J.H.P. prepared the materials, samples, heat treatment. L.L., J.H.P., and S.W. designed and conducted DTA. S.W. conducted the XRD and TEM experiments. S.W., M.W.Q., J.W., P.M.V., and I.S. prepared the manuscript, and all authors reviewed the manuscript.

## Competing interests

The authors declare no competing interests.

## Additional information

**Supplementary information** The online version contains show [QJ] [#,63]supplementary material available at https://doi.org/10.1038/s41467-026-69095-8.

