## [Transparent Peer Review file · Nature Communications]

Short-range order in high entropy carbides

Corresponding Author: Professor Izabela Szlufarska

Version 0:

Reviewer comments:

Reviewer #1

(Remarks to the Author)

After reviewing the manuscript, I regret to inform you that I cannot recommend it for publication in Nature Communications. My decision is based on several fundamental concerns regarding the methodology, data interpretation, and conclusions, which significantly impact the validity of the study.

A critical issue is the annealing process at 1773K for 24 hours. While this temperature might be reasonable for enhancing chemically short-range order (CSRO), the claim that the “as-synthesized sample” retains a CSRO structure characteristic of 300K contradicts fundamental thermodynamic expectations. The processing route described suggests that both the as-synthesized and aged samples should exhibit nearly the same CSRO levels, or at most a slight increase in the aged sample. Since the paper is largely built on this premise, the conclusions drawn from it are, in my view, fundamentally flawed.

The STEM-HAADF evidence also raises concerns. Given that both samples should inherently have similar CSRO amounts, the observed contrast could be attributed to sample preparation artifacts. Furthermore, the TEM analysis used to support the conclusions presents additional issues. Even if one assumes a significant difference in CSRO levels between the two materials, the domains observed in the figures extend over several nanometers—far beyond what would be expected for CSRO, which is typically a first- and second-nearest-neighbor phenomenon. Additionally, considering the thickness of the TEM sample (likely over 100 atoms), each “atom” in the images represents an average over a significant number of atomic sites. This averaging effect further invalidates the interpretation that these images provide direct evidence of CSRO. For these reasons, and possibly others, the imaging approach used in this study—though seen in the literature—has been contested in multiple prior publications.

Finally other concerns are the choice of compositions, which lacks a clear justification. Additionally, the atomistic simulations presented are too brief, and it is unclear whether sufficient time at 300K was allowed to capture potential phase formations. Given the likelihood of other long-range order (LRO) phases appearing at lower temperatures, a more comprehensive analysis—including WC parameters up to at least the 5th neighbor for all elements—would be necessary.

Given these fundamental flaws in the methodology and analysis, I do not believe that the manuscript meets the necessary standards for publication in Nature Communications.

Reviewer #2

(Remarks to the Author)

This paper uses a combination of atomistic modeling and experimental analysis to consider the existence and impact of chemical short range order (CSRO) in high entropy carbides. The results are interesting, though as detailed below I do have some questions about the experimental analysis. Further, these results are not overly surprising, given the same behavior has been seen in HEAs and HEOs. That the same behavior occurs in HECs is almost expected and while the confirmation of that behavior is valuable, I am not convinced it rises to the level of impact of Nature Communications. I would ask the authors to make a stronger case for why Nature Communications is appropriate for this work.

The abstract and paper note that CSRO has not been characterized in HECs and that they are special because only one sublattice exhibits mixing, but this is true of the disordered oxides and HEOs that the paper mentions. Why are the HECs any different in this regard? Shouldn't the concepts of CSRO in HEOs extend straight forwardly to HECs?

There is extensive work on short range order in disordered oxides by Maik Lang's group at Tennessee which would be relevant for the discussion here.

Why would 300K MD+MC simulations represent as-synthesized conditions? If anything, as-synthesized would be more non-equilibrium and further away from the 0K ground state, which the 300K simulations would be closer to.

Related to this, I'm a little confused as to the synthesis conditions. The heat treated sample was held at 1773 K for 24 hours, but the as-synthesized material was done at 2300 C (should use same units everywhere) so 2600 K - it seems that the as-synthesized material should have less SRO than the heat treated as it was not annealed at lower temperatures. I'm not sure what I'm missing.

Around line 89, where the authors say that CSRO is measured by the tendency of atoms to cluster, are they referring to short ranged clustering? In the SRO literature, there is a distinction between ordering and clustering and I just want to clarify which is meant here (both would be considered more ordered than random).

It isn't obvious to me that Z-contrast TEM or 4D STEM can reveal the SRO in these materials as the ordering would have to occur on length scales longer than the thickness of the sample. Wouldn't the resulting images be a convolution of the structure through the thickness? The foil thickness seems to be about 50 nm or so, so I might expect the images to be averaged over that thickness, making it nearly impossible to resolve features with length scales smaller than that. Can the authors comment on this?

The authors say that the domains in the simulations are likely smaller than in experiment because of the supercell size. Have they tested that the distribution varies as the cell size is changed?

The radiation studies provide a nice complement to the thermodynamic studies, though similar studies have been done in the past so these are not entirely novel. The authors conclude that, for a sample where presumably the only difference is CSRO, the sample with greater CSRO exhibits less swelling and smaller defects. Radiation is known to induce disordering in chemically complex systems - would this effect persist or would it saturate at some point as radiation drives the system to a more disordered/random state?

As the ion beam irradiations used Si as the ion species, could there be any chemical effect due to introducing Si?

Reviewer #3

(Remarks to the Author)

I have reviewed the manuscript and, unfortunately, I cannot support it being published in Nature Communications in its current form. I base this on several concerns I had with the paper, and these are summarized in the following points:

1. The synthesis of the HECs involves hot pressing at 2300°C and cooled slowly in the furnace to room temperature. If CSRO occurs upon cooling, it is unclear how the Mo sample heat treated at 1500°C and furnace cooled would be any different. In fact, it might be argued that it would potentially have greater CSRO rather than less. This is a fundamental issue to the paper given that the author's propose that the annealing should produce less SRO.
2. Given that CSRO is expected to exist over very small distances (a few atoms) and that the sample thicknesses are probably in the range of 100 atoms thick (the authors indicate similar thicknesses for their samples based on their EELS data but fail to provide actual numbers so this is an estimate of typical thin specimen dimensions), it is difficult to prove that the subtle differences in contrast result from CSRO – one could argue several other reasons for such contrast differences and it seems virtually impossible to say with certainty that these are due to CSRO.
3. Normally, ordering in alloys occurs when unlike atoms have a greater tendency to bond than like atoms due to a negative enthalpy of mixing of the unlike atoms. On the other hand, clustering of like atoms tends to reflect a positive enthalpy of mixing of unlike elements. Given that much of the argument is that like atoms (e.g., V-V and Zr-Zr) tend to cluster seems contrary to the idea of CSRO. Granted there are some cases where they do predict unlike atoms with greater tendency to cluster, but the premise is that SRO is greater in the HEC-Zr material is greater than in HEC-Mo where there is a tendency for unlike atoms to cluster according to the MC/MD simulations. Also, I suppose one can argue that in multicomponent alloys and carbides, it's likely you'll have atom pairs that have a tendency to order and pairs that have a tendency to cluster.
4. The radiation studies also seem somewhat flawed. Related to the first point above, it is not at all clear why the annealed HEC-Mo would behave differently and yet the authors conclude that the difference in volume changes (lattice expansion) due to irradiation is clear evidence of CSRO. Since the HEC-Zr with the largest purported CSRO has greater volume change, it is hard to prove that CSRO is playing a significant role in the different materials' resistances to radiation damage.

In summary, these rather serious concerns with the interpretation of the experimental data lead me to conclude that this paper should ready to be published in its current form.

Version 1:

Reviewer comments:

Reviewer #1

(Remarks to the Author)

The topic of the manuscript is relevant, and the results have the potential to be impactful. However, I believe that the main and most critical evidence supporting the claim that CSRO is a key parameter in this system is not convincingly demonstrated in the current version of the paper. I provide comments above to justify my reasoning and offer suggestions to the authors.

I remain seriously concerned about the actual difference between the two processing routes aimed at achieving distinct CSRO levels. Both 1573 K and 1773 K are high temperatures, and recent publications have shown that, at such conditions, the CSRO equilibrium state can be reached extremely quickly — sometimes in less than a second. Therefore, it is unclear whether these processing routes can truly produce different CSRO levels, even with water quenching. Strong and convincing experimental evidence distinguishing the two samples is, in my opinion, crucial to support any assertive claims regarding differences in CSRO. For reference, see the TTT diagram presented in DOI: 10.1016/j.actamat.2022.118314.

Furthermore, the HRTEM images remain unconvincing to me. Several classical studies have demonstrated the challenges of detecting CSRO via HRTEM and highlighted how the observed contrast may arise from various artifacts (see, for example, 10.1016/0304-3991(95)00040-8 and 10.1016/S1359-6454(98)00180-3). Therefore, I believe that, for this paper to be suitable for publication, it must include complementary evidence to validate potential differences in CSRO levels. There is a growing body of work using calorimetry, dilatometry, and electrical resistivity for such analyses. If the authors could demonstrate consistent results from one of these techniques that align with the expected trends for different CSRO levels, the claims in the paper would be much stronger and its impact significantly enhanced. The 4-D STEM and strain mapping is to my knowledge not yet well consolidated for this end, especially due to the small domain sizes compared to much larger sample thicknesses, same idea as the classical papers listed above criticize HRTEM for this end.

The experimental condition for CSRO is 1573 K. While it is reasonable for the authors to perform simulations at 1773 K and 300 K, it does not make sense to me to compare the latter directly with the experimental values. Why not simulate at 1573 K as well, or even at multiple temperatures, and then plot W–C as a function of temperature?

The presentation of Figure 1 is also not ideal for visualizing the W–C parameters. I suggest improving its layout for better clarity.

Reviewer #2

(Remarks to the Author)

The authors have done an admirable job of responding to my comments. (which were essentially the same as the other referees). While I still have some questions about interpretation, my concerns about the actual results have been more than adequately addressed, so I recommend publication.

There were three primary issues raised:

- The thought that the higher temperature anneals had more ordering than the lower temperature ones, which made no thermodynamic sense.
- The issue of the TEM averaging over regions bigger than the size of the CSRO domains.
- That CSRO in HECs is not so fundamentally different than in other ceramics.

The authors have addressed most of these very well. I'm overall satisfied with the response of the first two points. I would only say that

- Lang describes his results as a transformation to a new structure, but it is in the end too short range to be a true new structure and is (at least in my opinion) a manifestation of CSRO and nothing more.
- I don't understand why, if there is no enthalpic driving force for ordering in HECs, why would they order at all?
- If the reason the atomistics and experiment disagree on the size of the domains is because of larger range lattice relaxation, why wouldn't that also be captured in the simulations?

As for the last original point, about comparing CSRO in HECs and e.g. HEOs, I am still less convinced. Even if I agree that Lang's result is qualitatively different, there are literally decades of work on complex oxides such as spinels and pyrochlores showing the evolution of chemical disorder - which is a form of CSRO - in these materials under irradiation. Similarly, there are many papers that are related to inducing disorder/CSRO by other means, such as thermally and chemically. It is not so clear that there is any fundamental difference. I do agree that they are not identical, but it isn't clear that this is qualitatively different or just quantitatively different.

That said, I do believe the authors have done a very heroic job in responding to the previous reviews. My thoughts above are optional for the authors to consider, perhaps as future directions. These concerns are more about interpretation than about

the quality of the work, which I do think is good and has improved substantially since the last version.

Reviewer #3

(Remarks to the Author)

The authors have responded to the comments of the three reviewers where similar concerns were expressed, and I commend them for their efforts to clarify things. That said, I still have a few issues with the results that I would at least like to mention. I'll leave the decision to the editor as to whether they rise to the level of requiring further revision.

1. I am still not convinced that the HAADF images represent CSRO although I am not as fluent in the literature as the authors who argue that they are convinced of the interpretation of the subtle contrast differences.
2. Except for the Mo-C system, all the binary carbides have considerable solubility ranges up to 50%C and can be rich in the metallic component. The SEM images in the supplemental section suggest that the authors might have exceeded 50% C in the Zr-HEC sample, where graphite is clearly observed, whereas that does not appear to be the case in the Mo-HEC samples, at least from the data shown. When $C < 50\%$, the variation from stoichiometry results in constitutional vacancies on the C sublattice and I wonder if there is any influence of this on the resulting CSRO and the observed irradiation effects?
3. Somewhat related to this, the GIXRD data suggest that the lattice parameters in the unirradiated Mo-HEC materials are somewhat different, i.e., the material annealed at 1773 K appears to have a smaller lattice parameter than the as-synthesized material that presumably has CSRO according to their interpretation of the results. However, after irradiation at 300 and 600 °C, the peaks for the two Mo-HEC conditions approximately overlap. Thus, could one argue that they samples contain the same amount of damage after irradiation although the expansion is greater for the sample annealed at 1773 (without CSRO) for some reason?
4. The authors added a comment that partial melting might have occurred during synthesis. While that can't be ruled out, it seems unlikely based on the SEM observations as it should be possible to see evidence of any incipient melting in the microstructure. The authors might want to reconsider using this explanation.
5. I guess a final note is that it's unclear to me why the authors focused on the Mo-HEC instead of the Zr-HEC given their conclusion that the Zr-HEC has a greater tendency for CSRO. I would have thought that they would have conducted the heat treatments to influence "disordering" and to follow the irradiation damage on the Zr-HEC.

Version 2:

Reviewer comments:

Reviewer #1

(Remarks to the Author)

After carefully re-evaluating the manuscript and the supporting information, I have two remaining points:

First, the central narrative of the paper relies heavily on the claim that the authors have two distinct experimental states for the Mo-containing carbide: one with CSRO and one without CSRO. This distinction is essential for their interpretation of the radiation-damage results as well as for the broader implications they draw about tunability of CSRO. However, the manuscript does not present any experimental evidence that actually confirms the existence of a "zero-CSRO" state. All experimental studies I am aware of report that increasing temperature reduces CSRO but does not eliminate it entirely; in practice, full randomization is extremely difficult to achieve (if not impossible). Furthermore, there are also studies that show that even a very rapid cooling cannot avoid the CSRO formation during cooling at high temperatures. In the present manuscript, the calorimetry data seems to demonstrate that one processing condition has more CSRO than the other. This is not the same as demonstrating that the higher-temperature condition has no CSRO. On the contrary, based on the literature and on the authors' own data, I would expect measurable (though reduced) CSRO to remain. Quantifying this difference may indeed be challenging, and I agree that a qualitative comparison is still valuable, but presenting the lower-CSRO state as "fully disordered" is not supported by the data as written.

Second, regarding Figure S7, I do not see the evidence the authors claim to be present. If I understand the intention correctly, the authors are arguing that adding CSRO increases the diffuse background intensity in the CBED pattern, and that this increase can be used as an indicator of CSRO. However, many other factors produce the same effect, including specimen thickness, temperature and local crystal distortions unrelated to chemical order. CSRO is typically identified through specific diffuse-scattering features—extra maxima at particular reciprocal-space positions that correlate with ordering vectors—not via a general increase in background. Even the use of diffuse scattering for CSRO identification is non-trivial, as several papers have pointed out challenges in distinguishing CSRO-related diffuse features from other intrinsic and extrinsic contributions. Given these considerations, I do not find that Figure S7 meaningfully supports the manuscript's claims. The figure does not provide clear or interpretable evidence of CSRO, and, as currently presented, does not advance the narrative.

Reviewer #3

(Remarks to the Author)

I have reviewed the revised paper on high entropy carbides and feel that the authors have adequately addressed my concerns, which were similar to the concerns of the other reviewers. However, their modifications A few minor comments:

1. The authors incorrectly refer to an order-disorder transition in their discussion of their thermal analysis data. The classic description of order-disorder transitions involves going from a fully disordered phase to a fully long-range ordered phase either by a first-order or a second-order transition. In the HEC materials, the structure is already ordered but with the possibility that the atoms on the cation sites are not completely random and undergo CSRO. Thus, the authors should modify their description accordingly, i.e., they should refrain from referring to their CSRO structure as resulting from an ordering transition given the structure does not change crystal structure.

2. The authors refer to Fig. S11c in the text to represent partial (incipient) melting – it should be Fig. S12c.

Version 3:

Reviewer comments:

Reviewer #1

(Remarks to the Author)

The authors have address my comments adequately.

Point by point response to reviewer's comments

Reviewer #1:

After reviewing the manuscript, I regret to inform you that I cannot recommend it for publication in Nature Communications. My decision is based on several fundamental concerns regarding the methodology, data interpretation, and conclusions, which significantly impact the validity of the study.

Comment 1:

A critical issue is the annealing process at 1773K for 24 hours. While this temperature might be reasonable for enhancing chemically short-range order (CSRO), the claim that the “as-synthesized sample” retains a CSRO structure characteristic of 300K contradicts fundamental thermodynamic expectations. The processing route described suggests that both the as-synthesized and aged samples should exhibit nearly the same CSRO levels, or at most a slight increase in the aged sample. Since the paper is largely built on this premise, the conclusions drawn from it are, in my view, fundamentally flawed.

Thank you for your insightful comments regarding the potential discrepancy between our simulation predictions and experimental observations. We agree that the origin of CSRO in the as-synthesized samples is not the same as in our simulations at 300 K as the as-synthesized samples are quenched from high temperature where partial melts may form, and they may not reach fully homogenous states. The CSRO observed in as-synthesized samples are likely to originate from the solidification process as pointed out by *Han Y. et al. Nature Communications vol. 15, p.6486, 2024* . The goal of our original manuscript was to compare samples with varying degrees of CSRO, not necessarily their temperature dependence. We fully agree with the reviewer that the as-synthesized sample was not appropriate to conduct studies of temperature dependence and that the way the sample was introduced in the original manuscript could have led to confusion.

We have now conducted additional experiments to test the predictions of temperature dependence of CSRO, and our experiments are fully consistent with predictions from simulations. Specifically, we further annealed as-synthesized HEC-Mo at 1573 K for 100 hours, which led to homogenization of the sample. HRSTEM performed on the newly annealed HEC-Mo confirm that CSRO still exists (see Fig. 2c in revised manuscript). Considering that the CSRO almost disappears in HEC-Mo after annealing at 1773 K (Fig. 2d in the main text), it is clear that there is a transition from ordering to disordering in HEC-Mo with the increase of annealing temperature. This observation agrees very well with predictions of our simulations.

We have included the additional results in the revised manuscript.

Comment 2:

The STEM-HAADF evidence also raises concerns. Given that both samples should inherently have similar CSRO amounts, the observed contrast could be attributed to sample preparation artifacts. Furthermore, the TEM analysis used to support the conclusions presents additional issues. Even if one assumes a significant difference in CSRO levels between the two materials, the domains observed in the figures extend over several nanometers—far beyond what would be expected for CSRO, which is typically a first- and second-nearest-neighbor phenomenon. Additionally, considering the thickness of the TEM sample (likely over 100 atoms), each “atom” in the images represents an average over a significant number of atomic sites. This averaging effect further invalidates the interpretation that these images provide direct evidence of CSRO. For these reasons, and possibly others, the imaging approach used in this study—though seen in the literature—has been contested in multiple prior publications.

Thank you for the question. We have good reasons to believe that the contrast variations observed in HAADF and 4D-STEM are not artifacts, and result from the presence of CSRO. We have also conducted additional experiments to provide even stronger evidence.

First of all, we do agree with the reviewer that the contrast variance observed in HAADF images might potentially arise from other sources such as the surface oxides. Though we have tried our best to control them, it is impossible to fully exclude all these factors. However, in addition to HAADF imaging, 4D-STEM analysis was also employed to study the CSRO, which solely depends on the diffraction disk positions. Thus, unlike HAADF imaging, the 4D-STEM analysis is robust against variation in thickness, and immune to the presence or thickness of surface oxides. The presence of CSRO and their variations among samples were confirmed by our 4D-STEM analysis, which is consistent with our HAADF imaging observations as well as predictions from our simulations. Thus, we believe the contrast variance observed in HAADF images originated from the CSRO instead of other sources.

Regarding the relatively large domain size of CSRO in our 4D-STEM observation, we believe there are two reasons for that. First, during our measurement, we employed an electron probe with a diameter of 6 Å, which resulted inevitably in convolution and blurring of the domain boundaries. Consequently, the measured domains are enlarged. Second, though CSRO regions are relatively small, the lattice distortion and strain induced by them can extend beyond the CSRO, which also contributes to the enlarged domains. However, one should note that even considering these two factors, the majority of the domains have size in the range of 1-2 nm, which is consistent with what is expected for CSRO. We have included this discussion in our revised manuscript for completeness.

We fully agree with the reviewer that in principle there might be an averaging effect in STEM analysis and that one needs to be careful interpreting the results. We have performed additional simulations to demonstrate that the averaging effect is insignificant in our studies and therefore our interpretation of CSRO remains valid. More specifically, during the collection of STEM data, dynamical scattering leads to electron channeling effect. For example, when imaging zone-axis oriented crystals, the data does not reflect a simple two-dimensional projection of the structure. Instead, the on-column probe exhibits an intensity peak near the electron entrance surface (top surface), which means that atoms in a thin region near the top surface dominate the collected signals (see e.g., *Voyley P. M. et al. Microscopy and Microanalysis vol. 10, p. 291-300, 2004*). We performed simulations of electron beam intensity along a column of atoms with a convergence semiangle of 23.4 mrad to simulate the condition of HAADF-STEM imaging (see Fig. S2a, b). Intensity peaks can be observed in the region from the top surface to a depth of 4 nm for both samples (Fig. S2a, b), suggesting atoms in the top 4 nm contribute the most to our collected data. Atoms in the region extending from 4 nm to 10 nm from the surface can make contributions, but their contributions dropped by 50%. Contributions from atoms beyond 10 nm can be neglected. The depth of 4 nm where the intensity reaches the peak is comparable to the depth of field (5.9 nm) calculated by *Li L. et al. Nature Communications vol. 14, p.7448, 2023*, where HAADF-STEM was also employed to directly derive cation distributions in cation-disordered oxides. In summary, the atoms in the top 4 nm dominated the collected data during HAADF-STEM and the averaging effects are most significant in the top 4 nms. Considering that the majority of CSRO domains are in the range of 1-2 nm, they can be resolved by HAADF-STEM.

Regarding the 4D-STEM, a smaller convergence angle (3 mrad) was used during the data collection. The electron channeling effect under such circumstances was also explored (see Fig. S2c, d). Intensity peaks appear in the regions from the top surface to a depth of 8 nm for both samples (Fig. S2c, d), indicating atoms in the top 8 nm contribute the most to our collected data. Atoms in the region extending from 8 nm to 18 nm can make contributions, but their contributions decreased by 60%. Contributions from atoms below 18 nm can be neglected. Overall, atoms in the top 8 nm dominated the collected data during 4D-STEM and the averaging effects are most considerable in the top 8 nm. To explore if the CSRO-induced strain can still be resolved when the signals are averaged by the atoms in a relatively larger region, 4D-STEM analysis were performed on our MD/MC samples with a thickness of 7 nm, which is close to the dominant region of 8 nm from top surface. According to our analysis, the highest strains measured are 6.1%, 4.6% and 4.4% for HEC-Zr, HEC-Mo and HEC-Mo annealed at 1773 K, respectively. The trend of the highest strains among different samples agrees well with our experimentally measured strain trend calculated directly from 4D-STEM, illustrating that the strain induced by CSRO can be resolved by 4D-STEM and the averaging effects in the top 8 nm can be neglected. We have included this discussion in our revised manuscript for completeness.

Comment 3:

Finally other concerns are the choice of compositions, which lacks a clear justification. Additionally, the atomistic simulations presented are too brief, and it is unclear whether sufficient time at 300K was allowed to capture potential phase formations. Given the likelihood of other long-range order (LRO) phases appearing at lower temperatures, a more comprehensive analysis—including WC parameters up to at least the 5th neighbor for all elements—would be necessary.

Thank you for your suggestions. In the revised manuscript, we have provided a more extended justification for the choice of the compositions we studied.

Regarding the simulation time, we have calculated the potential energy of the systems and plotted it as a function of the number of iterations (see Fig. S1). According to our simulation, the energy of the simulated systems has converged, indicating an equilibrium state. Thus, we believe the simulation time is long enough to capture potential phase formation, if any. In addition, we do not expect any phase separation in the HECs studied by this work, as no second phase was observed in the XRD results (Fig. S9a). We have updated our manuscript to clarify this point.

Regarding the long-range order (LRO), we have included the Warren-Cowley parameter up to 3rd NN in Fig. 1 to exclude the formation of LRO. As shown in Fig. 1, at the 3rd NN, the atom arrangement in HEC-Zr and HEC-Mo at low temperature is almost random, indicating weak or even no ordering at and beyond the 3rd NN. Thus, we believe there is no LRO formed in our samples at low temperature. We have updated our manuscript to include this information.

Reviewer #2:

This paper uses a combination of atomistic modeling and experimental analysis to consider the existence and impact of chemical short range order (CSRO) in high entropy carbides. The results are interesting, though as detailed below I do have some questions about the experimental analysis. Further, these results are not overly surprising, given the same behavior has been seen in HEAs and HEOs. That the same behavior occurs in HECs is almost expected and while the confirmation of that behavior is valuable, I am not convinced it rises to the level of impact of Nature Communications. I would ask the authors to make a stronger case for why Nature Communications is appropriate for this work.

Comment 1:

The abstract and paper note that CSRO has not been characterized in HECs and that they are special because only one sublattice exhibits mixing, but this is true of the disordered oxides and HEOs that the paper mentions. Why are the HECs any different in this regard? Shouldn't the concepts of CSRO in HEOs extend straight forwardly to HECs? There is extensive work on short range order in disordered oxides by Maik Lang's group at Tennessee which would be relevant for the discussion here.

Thank you for your comments. We agree that the mixing on one sublattice is the same for HECs, disordered oxides, and HEOs. Instead, the fundamental difference between them is the bonding nature. In HECs, the covalent bonding between transition metal (TM) and carbon is more uniform and less selective, whereas in HEOs, the ionic bonding between TM and oxygen is less uniform and more selective. More specifically, different TMs exhibit varying oxidation states and electronic configurations in TM-O bonding, resulting in selective arrangement of TMs around oxygen. In other words, there is an enthalpic driving force for the formation of CSRO in oxides. However, in HECs, the differences in TM-C bonding among various TMs are minimal, offering negligible enthalpic driving force for CSRO. Thus, it would be difficult to extend the concept of CSRO in oxides directly to carbides. In addition, the SRO studied by Lang's group (e.g., Fuentes A. F. *et al. Applied Physics Reviews* vol. 11, 2024 Shamblin J. *Nature Materials* vol. 15, p. 507-511, 2016), involves formation of a new phase in a disordered oxide. The formation of a new phase with the size range of one to two unit cells in an disordered oxide can be referred to as SRO, but it is not

the same as formation of a local CSRO in a disordered phase without phase transformation. More specifically, in our work on HECs, CSRO is defined as the deviation of chemical species from random arrangement in the same sublattice without changing the structure. Thus, the SRO studied by Lang's group cannot be directly compared to the CSRO we focus on in HECs.

We have revised our manuscript to emphasize the differences among HECs, HEOs or disordered oxides in background and added references to Lang's work on oxides.

Comment 2:

Why would 300K MD+MC simulations represent as-synthesized conditions? If anything, as-synthesized would be more non-equilibrium and further away from the 0K ground state, which the 300K simulations would be closer to.

Related to this, I'm a little confused as to the synthesis conditions. The heat treated sample was held at 1773 K for 24 hours, but the as-synthesized material was done at 2300 C (should use same units everywhere) so 2600 K - it seems that the as-synthesized material should have less SRO than the heat treated as it was not annealed at lower temperatures. I'm not sure what I'm missing.

The goal of our original manuscript was to compare samples with varying degrees of CSRO, not necessarily the temperature dependence of CSRO and in that sense the 300K in simulations represented the as-synthesized sample in experiment. We agree, however, that this comparison can be confusing because the origin of CSRO in the as-synthesized samples is not the same as in our simulations at 300 K.

The as-synthesized samples are quenched from high temperature where partial melts may form, and they may not reach fully homogenous states. The CSRO observed in as-synthesized samples are likely to originate from the solidification process as pointed out by *Han Y. et al. Nature Communications vol. 15, p.6486, 2024*. Meanwhile, as illustrated by *Han Y. et al. Nature Communications vol. 15, p.6486, 2024*, CSRO with different origins, whether from quenching or annealing, exhibit similar chemical preferences. Thus, in the revised manuscript, CSRO observed in simulations at 300 K serves as a reference for the analysis of CSRO in experimental results.

To address the valid question about temperature dependence. we have now conducted additional experiments to test predictions from our simulations. Specifically, we further annealed as-synthesized HEC-Mo at 1573 K for 100 hours, which led to homogenization of the sample. HRSTEM performed on the newly annealed HEC-Mo confirm that CSRO still exists (see Fig. 2c in revised manuscript). Considering that the CSRO almost disappears in HEC-Mo after annealing at 1773 K (Fig. 2d in the main text), it is clear that there is a transition from ordering to disordering in HEC-Mo with the increase of annealing temperature. This observation agrees very well with predictions of our simulations.

We have included the additional results in the revised manuscript.

Comment 3:

Around line 89, where the authors say that CSRO is measured by the tendency of atoms to cluster, are they referring to short ranged clustering? In the SRO literature, there is a distinction between ordering and clustering and I just want to clarify which is meant here (both would be considered more ordered than random).

Thank you for your question. In previous work by *Abu-Odeh A. et al. Acta Materialia vol. 275, p. 119185, 2023*, two kinds of non-random correlations between different chemical species in disordered solid solution alloys were defined. When unlike species attract each other, short-range ordering (SRO) occurs and when like atoms attract each other, short-range clustering (SRC) occurs. In our work, the CSRO is defined as deviation of chemical species from random arrangement of cations on the same sublattice without changing the structure, which includes both SRO and SRC. We have revised our manuscript to clarify the definition of CSRO.

Comment 4:

It isn't obvious to me that Z-contrast TEM or 4D STEM can reveal the SRO in these materials as the ordering would have to occur on length scales longer than the thickness of the sample. Wouldn't the resulting images be a convolution of the structure through the thickness? The foil thickness seems to be about 50 nm or so, so I might expect the images to be averaged over that thickness, making it nearly impossible to resolve features with length scales smaller than that. Can the authors comment on this?

We fully agree with the reviewer that in principle there might be an averaging effect in STEM analysis and that one needs to be careful interpreting the results. We have performed additional simulations to demonstrate that the averaging effect is insignificant in our studies and therefore our interpretation of CSRO remains valid. More specifically, during the collection of STEM data, dynamical scattering leads to electron channeling effect. For example, when imaging zone-axis oriented crystals, the data does not reflect a simple two-dimensional projection of the structure. Instead, the on-column probe exhibits an intensity peak near the electron entrance surface (top surface), which means that atoms in a thin region near the top surface dominate the collected signals (see e.g., *Voyles P. M. et al. Microscopy and Microanalysis vol. 10, p. 291-300, 2004*). We performed simulations of electron beam intensity along a column of atoms with a convergence semiangle of 23.4 mrad to simulate the condition of HAADF-STEM imaging (see Fig. S2a, b). Intensity peaks can be observed in the region from the top surface to a depth of 4 nm for both samples (Fig. S2a, b), suggesting atoms in the top 4 nm contribute the most to our collected data. Atoms in the region extending from 4 nm to 10 nm from the surface can make contributions, but their contributions dropped by 50%. Contributions from atoms beyond 10 nm can be neglected. The depth of 4 nm where the intensity reaches the peak is comparable to the depth of field (5.9 nm) calculated by *Li L. et al. Nature Communications vol. 14, p.7448, 2023*, where HAADF-STEM was also employed to directly derive cation distributions in cation-disordered oxides. In summary, the atoms in the top 4 nm dominated the collected data during HAADF-STEM. And the averaging effects are most significant in the top 4 nms. Considering that the majority of CSRO domains are in the range of 1-2 nm, they can be resolved by HAADF-STEM.

Regarding the 4D-STEM, a smaller convergence angle (3 mrad) was used during the data collection. The electron channeling effect under such circumstances was also explored (see Fig. S2c, d). Intensity peaks appear in the regions from the top surface to a depth of 8 nm for both samples (Fig. S2c, d), indicating atoms in the top 8 nm contribute the most to our collected data. Atoms in the region extending from 8 nm to 18 nm can make contributions, but their contributions decreased by 60 %. Contributions from atoms below 18 nm can be neglected. Overall, atoms in the top 8 nm dominated the collected data during 4D-STEM and the averaging effects are most considerable in the top 8 nm. To explore if the CSRO induced strain can still be resolved when the signals are averaged by the atoms in a relatively larger region, 4D-STEM analysis were performed on our MD/MC samples with a thickness of 7 nm, which is close to the dominant region of 8 nm from top surface. According to our analysis, the highest strains measured are 6.1%, 4.6% and 4.4% for HEC-Zr, HEC-Mo and HEC-Mo annealed at 1773 K, respectively. The trend of the highest strains among different samples agrees well with our experimentally measured strain trend calculated directly from 4D-STEM, illustrating that the strain induced by CSRO can be resolved by 4D-STEM and the averaging effects in the top 8 nm can be neglected.

In the revised manuscript we have included the measurements of sample thickness, which were carried out using two methods. First, we completed the measurement of thickness using EELS log-ratio method. By calculating the inelastic mean free path of HEC-Zr (82.9 nm) and HEC-Mo (82.6 nm) using the model proposed by *Malis T. et al. Journal of electron microscopy technique vol. 8, p. 193-200, 1988* and *Egerton R. F. et al. Ultramicroscopy vol. 21, p. 231-244, 1987*, the thickness of HEC-Zr, HEC-Mo, HEC-Mo annealed at 1573 K and HEC-Mo annealed at 1773 K were calculated to be 41.4 nm, 39.6 nm, 29.7 nm, and 37.2 nm, respectively. The measurements are complemented by the collected diffraction pattern of 4D-STEM. More specifically, the ratio of transmitted beam intensity to a {111} diffraction beam intensity is dependent on dynamical scattering, which is related to sample thickness. Compared to the beam intensity ratios calculated from MD/MC models with different thickness, the thickness of HEC-Zr, HEC-Mo, HEC-

Mo annealed at 1573 K and HEC-Mo annealed at 1773 K are measured to be 41.4 nm, 39.6 nm, 29.7 nm and 37.2 nm, respectively. The thickness measured by two different methods is reasonably comparable to each other.

Comment 5:

The authors say that the domains in the simulations are likely smaller than in experiment because of the supercell size. Have they tested that the distribution varies as the cell size is changed?

Thank you for your helpful comment. This statement was a speculation, and we agree that it should be tested. To clarify this statement, we performed an additional MD/MC simulation on HEC-Mo with a larger supercell (32,768 atoms, compared to 21,952 in the original simulations) at 300 K. According to the WC parameters, the CSRO in both supercells are comparable. Thus, the size of the supercell is not the reason for smaller domain size in simulations than in experiments. We have revised the manuscript and no longer included this speculation.

Regarding the difference in domain size between simulation and experimental observations, we believe there are two possible explanations. First, during the measurement of STEM, an electron probe with a diameter of 6 Å was employed. Therefore, the domain boundaries are inevitably convolved and blurred, resulting in the enlargement of CSRO domains. Second, though CSRO regions are relatively small, the lattice distortion and strain induced by them can extend beyond the CSRO, which also contributes to the enlarged domains. We have updated the manuscript to discuss the possible reasons for the larger domains observed in experiments than in simulations.

Comment 6:

The radiation studies provide a nice complement to the thermodynamic studies, though similar studies have been done in the past so these are not entirely novel. The authors conclude that, for a sample where presumably the only difference is CSRO, the sample with greater CSRO exhibits less swelling and smaller defects. Radiation is known to induce disordering in chemically complex systems - would this effect persist or would it saturate at some point as radiation drives the system to a more disordered/random state?

Thank you for bringing the evolution of CSRO during radiation to our attention, which is indeed interesting and worth exploring. However, such study is beyond the scope of this work, and we have updated the manuscript to include a brief discussion posing this interesting question that emerges from these results.

Comment 7:

As the ion beam irradiations used Si as the ion species, could there be any chemical effect due to introducing Si?

Thank you for your question. The effect of introducing Si on chemical ordering is negligible. More specifically, we have calculated Si concentration as a function of depth from the surface using SRIM (see Fig. S6b). The maximum concentrations of Si in both HECs are less than 0.8 %, and therefore its impact on variation in the local chemistry is negligible. We have included this discussion in our revised manuscript.

Reviewer #3:

I have reviewed the manuscript and, unfortunately, I cannot support it being published in Nature Communications in its current form. I base this on several concerns I had with the paper, and these are summarized in the following points:

Comment 1:

The synthesis of the HECs involves hot pressing at 2300°C and cooled slowly in the furnace to room temperature. If CSRO occurs upon cooling, it is unclear how the Mo sample heat treated at 1500°C and furnace cooled would be any different. In fact, it might be argued that it would potentially have greater CSRO rather than less. This is a fundamental issue to the paper given that the author's propose that the annealing should produce less SRO.

Thank you for your insightful comments regarding the potential discrepancy between our simulation predictions and experimental observations. We agree that the origin of CSRO in the as-synthesized samples is not the same as in our simulations at 300 K as the as-synthesized samples are quenched from high temperature where partial melts may form, and they may not reach fully homogenous states. The CSRO observed in as-synthesized samples are likely to originate from the solidification process as pointed out by *Han Y. et al. Nature Communications vol. 15, p.6486, 2024*. The goal of our original manuscript was to compare samples with varying degrees of CSRO, not necessarily their temperature dependence. We fully agree with the reviewer that the as-synthesized sample was not appropriate to conduct studies of temperature dependence and that the way the sample was introduced in the original manuscript could have led to confusion.

We have now conducted additional experiments to test the predictions of temperature dependence of CSRO and our experiments are fully consistent with predictions from simulations. Specifically, we further annealed as-synthesized HEC-Mo at 1573 K for 100 hours, which led to homogenization of the sample. HRSTEM performed on the newly annealed HEC-Mo confirm that CSRO still exists (see Fig. 2c in revised manuscript). Considering that the CSRO almost disappears in HEC-Mo after annealing at 1773 K (Fig. 2d in the main text), it is clear that there is a transition from ordering to disordering in HEC-Mo with the increase of annealing temperature. This observation agrees very well with predictions of our simulations.

We have included the additional results in the revised manuscript.

Comment 2:

Given that CSRO is expected to exist over very small distances (a few atoms) and that the sample thicknesses are probably in the range of 100 atoms thick (the authors indicate similar thicknesses for their samples based on their EELS data but fail to provide actual numbers so this is an estimate of typical thin specimen dimensions), it is difficult to prove that the subtle differences in contrast result from CSRO – one could argue several other reasons for such contrast differences and it seems virtually impossible to say with certainty that these are due to CSRO.

We have now included the measurements of sample's thickness in our revised manuscript. To measure sample thickness, we employed two methods that are complementary to each other. First, we completed the measurement of thickness using EELS log-ratio method. By calculating the inelastic mean free path of HEC-Zr (82.9 nm) and HEC-Mo (82.6 nm) using the model proposed by *Malis T. et al. Journal of electron microscopy technique vol. 8, p. 193-200, 1988* and *Egerton R. F. et al. Ultramicroscopy vol. 21, p. 231-244, 1987*, the thickness of HEC-Zr, HEC-Mo, HEC-Mo annealed at 1573 K and HEC-Mo annealed at 1773 K are calculated to be 41.4 nm, 39.6 nm, 29.7 nm and 37.2 nm, respectively. The measurements are complemented by the collected diffraction pattern of 4D-STEM. More specifically, the ratio of transmitted beam intensity to a {111} diffraction beam intensity is dependent on dynamical scattering, which is related to sample thickness. Compared to the beam intensity ratios calculated from MD/MC models with different thickness, the thickness of HEC-Zr, HEC-Mo, HEC-Mo annealed at 1573 K and HEC-Mo annealed at 1773 K are measured to be 41.4 nm, 39.6 nm, 29.7 nm and 37.2 nm, respectively. The thickness values measured by two different methods are in a reasonably agreement with each other. With respect to the contrast variations observed in HAADF and 4D-STEM, we have good reasons to believe they are not artifacts, and that the variations result from the presence of CSRO and we have conducted additional experiments to provide even stronger evidence.

First of all, we do agree with the reviewer that the contrast variance observed in HAADF images might potentially arise from other sources such as the surface oxides. Though we have tried our best to control

them, it is impossible to fully exclude all these factors. However, in addition to HAADF imaging, 4D-STEM analysis was also employed to study the CSRO, which solely depends on the diffraction disk positions. Thus, unlike HAADF imaging, the 4D-STEM analysis is robust against variation in thickness, and immune to the presence or thickness of surface oxides. The presence of CSRO and their variations among samples were confirmed by our 4D-STEM analysis, which is consistent with our HAADF imaging observations as well as predictions from our simulations. Thus, we believe the contrast variance observed in HAADF images originated from the CSRO instead of other sources.

We fully agree with the reviewer that in principle there might be an averaging effect in STEM analysis and that one needs to be careful interpreting the results. We have performed additional simulations to demonstrate that the averaging effect is insignificant in our studies and therefore our interpretation of CSRO remains valid. More specifically, during the collection of STEM data, dynamical scattering leads to electron channeling effect. For example, when imaging zone-axis oriented crystals, the data does not reflect a simple two-dimensional projection of the structure. Instead, the on-column probe exhibits an intensity peak near the electron entrance surface (top surface), which means that atoms in a thin region near the top surface dominate the collected signals (see e.g., *Voyle P. M. et al. Microscopy and Microanalysis vol. 10, p. 291-300, 2004*). We performed simulations of electron beam intensity along a column of atoms with a convergence semiangle of 23.4 mrad to simulate the condition of HAADF-STEM imaging (see Fig. S2a, b). Intensity peaks can be observed in the region from the top surface to a depth of 4 nm for both samples (Fig. S2a, b), suggesting atoms in the top 4 nm contribute the most to our collected data. Atoms in the region extending from 4 nm to 10 nm from the surface can make contributions, but their contributions dropped by 50%. Contributions from atoms beyond 10 nm can be neglected. The depth of 4 nm where the intensity reaches the peak is comparable to the depth of field (5.9 nm) calculated by *Li L. et al. Nature Communications vol. 14, p.7448, 2023*, where HAADF-STEM was also employed to directly derive cation distributions in cation-disordered oxides. In summary, the atoms in the top 4 nm dominated the collected data during HAADF-STEM. And the averaging effects are most significant in the top 4 nms. Considering that the majority of CSRO domains are in the range of 1-2 nm, they can be resolved by HAADF-STEM.

Regarding the 4D-STEM, a smaller convergence angle (3 mrad) was used. The electron channeling effect under such circumstances was also explored (see Fig. S2c, d). Intensity peaks appear in the regions from the top surface to a depth of 8 nm for both samples (Fig. S2c, d), indicating atoms in the top 8 nm contribute the most to our collected data. Atoms in the region extending from 8 nm to 18 nm can make contributions, but their contributions decreased by 60 %. Contributions from atoms below 18 nm can be neglected. Overall, atoms in the top 8 nm dominated the collected data during 4D-STEM and the averaging effects are most considerable in the top 8 nm. To explore if the CSRO induced strain can still be resolved when the signals are averaged by the atoms in a relatively larger region, 4D-STEM analysis were performed on our MD/MC samples with a thickness of 7 nm, which is close to the dominant region of 8 nm from top surface. According to our analysis, the highest strains measured are 6.1%, 4.6% and 4.4% for HEC-Zr, HEC-Mo and HEC-Mo annealed at 1773 K, respectively. The trend in the highest strains among different samples agrees well with our experimentally measured strain trend calculated directly from 4D-STEM, illustrating that the strain induced by CSRO can be resolved by 4D-STEM and the averaging effects in the top 8 nm can be neglected.

Comment 3:

Normally, ordering in alloys occurs when unlike atoms have a greater tendency to bond than like atoms due to a negative enthalpy of mixing of the unlike atoms. On the other hand, clustering of like atoms tends to reflect a positive enthalpy of mixing of unlike elements. Given that much of the argument is that like atoms (e.g., V-V and Zr-Zr) tend to cluster seems contrary to the idea of CSRO. Granted there are some cases where they do predict unlike atoms with greater tendency to cluster, but the premise is that SRO is greater in the HEC-Zr material is greater than in HEC-Mo where there is a tendency for unlike atoms to cluster according to the MC/MD simulations. Also, I suppose one can argue that in multicomponent alloys and carbides, it's likely you'll have atom pairs that have a tendency to order and pairs that have a tendency to cluster.

We appreciate the reviewer's comments regarding the distinction between short-range ordering (SRO) and short-range clustering (SRC), and we agree that there are different ways to define the local ordering. To clarify our approach, we have updated the manuscript to define the CSRO as the deviation of chemical order from random arrangement on the same sublattice without changing the lattice structure, which includes both SRO and SRC. In our work, the degree of SRO in HEC-Zr and HEC-Mo is comparable. For example, in HEC-Zr, Nb-Ti and Nb-Zr are pairs with strong chemical preferences, with α_{ij} equal to -0.28 and -0.17, respectively. In HEC-Mo, Nb-Ti and Ti-Mo are pairs with strong chemical preferences, with α_{ij} equal to -0.17 and -0.26, respectively. Though SRO in two samples is comparable, the SRC in HEC-Zr is stronger than that in HEC-Mo. Thus, it is reasonable to conclude that the degree of CSRO in HEC-Zr is greater than that in HEC-Mo. We have updated our manuscript to clarify the definition of CSRO and to include this discussion of SRO and SRC.

Comment 4:

The radiation studies also seem somewhat flawed. Related to the first point above, it is not at all clear why the annealed HEC-Mo would behave differently and yet the authors conclude that the difference in volume changes (lattice expansion) due to irradiation is clear evidence of CSRO. Since the HEC-Zr with the largest purported CSRO has greater volume change, it is hard to prove that CSRO is playing a significant role in the different materials' resistances to radiation damage.

We would like to clarify that the difference in volume change of HEC-Mo samples due to radiation is the result of CSRO instead of being the evidence of CSRO. More specifically, we have prepared HEC-Mo samples with and without CSRO, which is HEC-Mo as-synthesized and HEC-Mo annealed at 1773 K, respectively. Both samples share the identical composition and structure, and the only difference is the degree of CSRO. Therefore, we attribute that the difference in radiation induced volume change to the existence of CSRO. However, studies on HEC-Zr suggest that the CSRO is not the sole factor that will impact the radiation induced damage. As noted in our original manuscript, the specific choice of chemical species also plays a role in determining radiation resistance. Therefore, though CSRO impacts the resistance to radiation induced damage, it is not the full picture and we do not claim that it is.

Point by point response to reviewer's comments

Reviewer #1:

The topic of the manuscript is relevant, and the results have the potential to be impactful. However, I believe that the main and most critical evidence supporting the claim that CSRO is a key parameter in this system is not convincingly demonstrated in the current version of the paper. I provide comments above to justify my reasoning and offer suggestions to the authors.

Comment 1:

I remain seriously concerned about the actual difference between the two processing routes aimed at achieving distinct CSRO levels. Both 1573 K and 1773 K are high temperatures, and recent publications have shown that, at such conditions, the CSRO equilibrium state can be reached extremely quickly — sometimes in less than a second. Therefore, it is unclear whether these processing routes can truly produce different CSRO levels, even with water quenching. Strong and convincing experimental evidence distinguishing the two samples is, in my opinion, crucial to support any assertive claims regarding differences in CSRO. For reference, see the TTT diagram presented in DOI: 10.1016/j.actamat.2022.118314.

Thank you for bringing to our attention the reference [J-P. Du et al. Acta Materialia 240 (2022) 118314], which discusses the formation of CSRO in CrCoNi. We agree that the TTT diagram serves as a useful guide in using thermal treatment to adjust the level of CSRO in materials. However, the TTT diagram of CrCoNi cannot be applied to HECs directly due to their distinct melting temperatures (T_m). More specifically, T_m of HEC-Mo is 3343 K and T_m of CrCoNi is 1530 K [J-P. Du et al. Acta Materialia 240 (2022) 118314]. Even at the same temperature, 1573 K or 1773 K for example, the atomic mobility in these two systems will be significantly different. These two temperatures correspond to $0.47 T_m$ and $0.53 T_m$ in HEC-Mo. The same homologous temperatures in CrCoNi would correspond to 720 K and 811 K, respectively. According to the TTT diagram from the above reference, the CSRO domain can reach $\sim 15 \text{ \AA}$ in diameter at 811 K but can grow up to $\sim 21 \text{ \AA}$ at 720 K. Thus, the degree of CSRO at these two conditions is meaningfully different. There is therefore no contradiction between our findings of varied degree of CSRO and the results reported by Du et al. In addition, we have carried out new DTA experiments on HEC-Mo, which suggests that the order to disorder transition temperature (T_d^{exp}) in this sample is $\sim 1653 \text{ K}$ (Fig. 2a and Fig. S3b). The fact that the transition temperature is between 1573 K and 1773 K further corroborates that our thermal treatment at these two conditions can result in different degrees of CSRO in HECs.

Comment 2:

Furthermore, the HRTEM images remain unconvincing to me. Several classical studies have demonstrated the challenges of detecting CSRO via HRTEM and highlighted how the observed contrast may arise from various artifacts (see, for example, 10.1016/0304-3991(95)00040-8 and 10.1016/S1359-6454(98)00180-3). Therefore, I believe that, for this paper to be suitable for publication, it must include complementary evidence to validate potential differences in CSRO levels. There is a growing body of work using calorimetry, dilatometry, and electrical resistivity for such analyses. If the authors could demonstrate consistent results from one of these techniques that align with the expected trends for different CSRO levels, the claims in the paper would be much stronger and its impact significantly enhanced. The 4-D STEM and strain mapping is to my knowledge not yet well consolidated for this end, especially due to the small domain sizes compared to much larger sample thicknesses, same idea as the classical papers listed above criticise HRTEM for this end.

Thank you for your valuable suggestions. We agree with the reviewer that complementary experiments can further strengthen our work, and we have performed differential thermal analysis (DTA) on HEC samples. The DTA curves confirm the existence of CSRO in our HECs. More specifically, for as-

synthesized samples, an exothermic peak emerges near 1473 K, indicating the formation of CSRO with the increase of temperature (Fig. 2a). A larger endothermic peak appears after the exothermic peak and ends at ~1653 K (Fig. 2a), indicating the dissolution of CSRO. The fact that the CSRO dissolution peak is larger than the formation peak indicates that there is pre-existing CSRO in as-synthesized HECs. However, it is difficult to compare HEC-Zr to HEC-Mo in terms of the amount of pre-existing CSRO due to the different ordering states of these materials. The order-disorder transition was also confirmed in HEC-Mo using DTA. More specifically, HEC-Mo was first annealed at 1573 K where ordering is preferred (Fig. S3b). The DTA curve upon heating exhibits a minimal CSRO formation peak followed by a significantly large dissolution peak (Fig. 2a), indicating that HEC-Mo annealed at 1573 K is ordered. HEC-Mo was further annealed at 1773 K, where disorder is preferred (Fig. S3b). DTA curve upon heating shows a large CSRO formation peak followed by a CSRO dissolution peak, which is slightly larger than the formation peak, suggesting that HEC-Mo annealed at 1773 K is disordered. In summary, our DTA provides additional strong evidence that CSRO exists in as-synthesized HECs and that order-disorder transition can occur in HECs with the change of temperature. These results are consistent with our simulations and STEM observations. Thus, they can serve as good complementary validation for our conclusion. We have included our DTA measurements in our revised manuscript.

Comment 3:

The experimental condition for CSRO is 1573 K. While it is reasonable for the authors to perform simulations at 1773 K and 300 K, it does not make sense to me to compare the latter directly with the experimental values. Why not simulate at 1573 K as well, or even at multiple temperatures, and then plot W–C as a function of temperature?

Thank you for your question. We have further performed MD/MC simulations on HEC-Mo at 600 K, 900 K, 1200 K, and 1573 K to understand the temperature dependence of CSRO. The WC parameters of HEC-Mo at different temperatures were plotted in Fig. S3a and it is clear the degree of CSRO decreases with increasing temperature. The temperature (T_d^{sim}) where order-to-disorder transition takes place is estimated to be ~900 K (Fig. S3a). This transition is lower than T_d^{exp} (~1653 K) measured using DTA (Fig. 2a and Fig. S3b). It is not uncommon for interatomic potentials to show deviations in predictions of transition temperatures and therefore based on our simulations we do not claim to make quantitative predictions of the extent of CSRO. We use the potential to predict the presence of CSRO and to show that CSRO decreases with increasing temperature. For this reason, we chose two simulation temperatures – one below and one above T_d^{sim} .

Although the reviewer did not ask this question, there is also a good reason why we did not perform annealing experiments at 300K, which would match the lower temperature used in simulations. The reason is that the kinetic temperature in experiments (based on DTA) is ~1473 K (see Fig. 2a, Fig. S3b), which means that below this temperature kinetics in the system is frozen and we would not be able to observe ordering on the time scales of the experiments.

The temperature of 1773K is above both T_d^{sim} and T_d^{exp} and therefore we were able to use this temperature in both simulations and experiments to achieve a fully disordered state.

In summary, because of the kinetic limitation in experiments and the mismatch in the order-disorder transition temperature between simulations and experiments, we chose different temperatures for simulations and experiments in the low-temperature regime to provide evidence and understanding of CSRO in HECs. We have included this discussion in our supplementary information.

Comment 4:

The presentation of Figure 1 is also not ideal for visualizing the W–C parameters. I suggest improving its layout for better clarity

Thank you for your suggestion. We have updated our Fig. 1 to only reflect the WC parameters of the 1st NN in both random systems and in our MD/MC simulated systems. The WC parameters of the 2nd and 3rd NN in our MD/MC simulated systems are moved to Fig. S2 for clarity.

Reviewer #2:

The authors have done an admirable job of responding to my comments. (which were essentially the same as the other referees). While I still have some questions about interpretation, my concerns about the actual results have been more than adequately addressed, so I recommend publication.

There were three primary issues raised:

- The thought that the higher temperature anneals had more ordering than the lower temperature ones, which made no thermodynamic sense.
- The issue of the TEM averaging over regions bigger than the size of the CSRO domains.
- That CSRO in HECs is not so fundamentally different than in other ceramics.

The authors have addressed most of these very well. I'm overall satisfied with the response of the first two points. I would only say that

- Lang describes his results as a transformation to a new structure, but it is in the end too short range to be a true new structure and is (at least in my opinion) a manifestation of CSRO and nothing more.
- I don't understand why, if there is no enthalpic driving force for ordering in HECs, why would they order at all?

Thank you for your question. In fact, there is enthalpic driving force for ordering in HECs, which is the formation enthalpy. More specifically, using the method described in our previous work [M. W. Qureshi et al *Materials Advances* 6 (2025) 5286-5294], the formation enthalpies of HEC-Zr and HEC-Mo are 68.78 meV/atom and 48.93 meV/atom, respectively. Thus, these two HECs are stabilized by entropy and the driving force for ordering is the formation enthalpy. We have included this information in our revised manuscript.

- If the reason the atomistics and experiment disagree on the size of the domains is because of larger range lattice relaxation, why wouldn't that also be captured in the simulations?

Thank you for your comments. In fact, there may exist multiple reasons for the discrepancy between the domain size of atomic simulations and the experiments, including the difference in the predicted order-disorder transition temperature. We agree that these reasons are speculations, and we made it clear in the revised manuscript.

As for the last original point, about comparing CSRO in HECs and e.g. HEOs, I am still less convinced. Even if I agree that Lang's result is qualitatively different, there are literally decades of work on complex oxides such as spinels and pyrochlores showing the evolution of chemical disorder - which is a form of CSRO - in these materials under irradiation. Similarly, there are many papers that are related to inducing disorder/CSRO by other means, such as thermally and chemically. It is not so clear that there is any fundamental difference. I do agree that they are not identical, but it isn't clear that this is qualitatively different or just quantitatively different.

That said, I do believe the authors have done a very heroic job in responding to the previous reviews. My thoughts above are optional for the authors to consider, perhaps as future directions. These concerns are

more about interpretation than about the quality of the work, which I do think is good and has improved substantially since the last version.

Thank you for your comments. We agree that there are multiple works discussing the ordering in complex oxides. However, HECs have very different bonding nature when compared to oxides, which is expected to result in unique defect kinetics and unique impact of CSRO on such kinetics. In fact, we have already performed atomistic simulations that reveal these effects. However, discussing details of defect kinetics in HECs is beyond the scope of the current paper, which is focused on the discovery and evidence for CSRO. Atomistic details of defect kinetics will be discussed in a separate paper.

Reviewer #3:

The authors have responded to the comments of the three reviewers where similar concerns were expressed, and I commend them for their efforts to clarify things. That said, I still have a few issues with the results that I would at least like to mention. I'll leave the decision to the editor as to whether they rise to the level of requiring further revision.

Comment 1:

I am still not convinced that the HAADF images represent CSRO although I am not as fluent in the literature as the authors who argue that they are convinced of the interpretation of the subtle contrast differences.

Thank you for your comments, as suggested by reviewer #1, we have performed DTA on HECs and they show consistent results with our simulations and STEM observation. These results further proved the reliability of our STEM observations.

Comment 2:

Except for the Mo-C system, all the binary carbides have considerable solubility ranges up to 50% C and can be rich in the metallic component. The SEM images in the supplemental section suggest that the authors might have exceeded 50% C in the Zr-HEC sample, where graphite is clearly observed, whereas that does not appear to be the case in the Mo-HEC samples, at least from the data shown. When $C < 50\%$, the variation from stoichiometry results in constitutional vacancies on the C sublattice and I wonder if there is any influence of this on the resulting CSRO and the observed irradiation effects?

Thank you for your comments. The reviewer is correct that HEC samples in this work are enriched in carbon. However, the effect of carbon enrichment on CSRO and observed irradiation resistance is minimal.

First, we want to clarify that there is no observable graphite in our samples. There are some black regions in the image of HEC-Zr. However, we have imaged these black regions using backscattered electron imaging and found that they are porosities rather than graphite particles (Fig. 1 in response letter). There may exist some graphite since our samples are C-rich, but their size and amount are much smaller than expected by the reviewer, as evidenced by no graphite peaks in XRD measurements (Fig. S11a).

Second, porosities have no impact on CSRO but can affect the radiation effects. For example, porosities can serve as defect sinks and accelerate the defect recovery in materials. In other words, porosities in HEC-Zr can suppress the defects accumulation and thus the lattice expansion. Considering that the HEC-Zr has more lattice expansion when compared to HEC-Mo even with more porosities, the effect of porosities in HEC-Zr on the radiation effect will not change our qualitative conclusion.

Third, when discussing the effect of CSRO on radiation resistance, we focus on HEC-Mo with and without CSRO because the samples share the same chemistry and differ only in the degree of CSRO. Thus, the effect of CSRO on radiation resistance discussed in our manuscript is not affected by C stoichiometry. We have also performed XPS measurements on HEC-Zr and HEC-Mo after Ar plasma cleaning for 100 s and the compositions are listed in Table S1. Based on our XPS measurements, the C concentration in HEC-

Zr is 56.72 at.% and the C concentration in HEC-Mo is 52.60 at.%, suggesting both HECs are C-rich. That means that the concentration of C vacancies in both samples is negligible. Regarding the effect of carbon off-stoichiometries on defect accumulation in carbides, previous study [Y. Yang et al. Journal of Nuclear Materials 454 (2014) 130-135] does suggest that the existence of graphite can serve as a strong source of knock-on carbon interstitials into carbides and lead to larger dislocation loops. However, our simulations suggested that the C interstitials in HECs share comparable migration energies (3.56 eV for HEC-Zr and 3.86 eV for HEC-Mo) and are immobile at the irradiation temperature. Thus, if graphite is present in these samples and introduces some C interstitials into HECs, these interstitials cannot migrate and coalesce to larger dislocation loops and the impact on lattice expansion is minimal.

In summary, the stoichiometry in HECs has minimal effect on CSRO and radiation resistance.

Figure 1: Backscattered electron imaging of HEC-Zr. (a) HEC-Zr sample with multiple pores. (b) Image of a pore within marked region in (a).

Comment 3:

Somewhat related to this, the GIXRD data suggest that the lattice parameters in the unirradiated Mo-HEC materials are somewhat different, i.e., the material annealed at 1773 K appears to have a smaller lattice parameter than the as-synthesized material that presumably has CSRO according to their interpretation of the results. However, after irradiation at 300 and 600 °C, the peaks for the two Mo-HEC conditions approximately overlap. Thus, could one argue that they samples contain the same amount of damage after irradiation although the expansion is greater for the sample annealed at 1773 (without CSRO) for some reason?

Thank you for your question. The reason for using the change in the lattice parameter instead of using the lattice parameter itself is that the accumulated defects in sample can lead to lattice expansion. More specifically, when defects accumulate in the material after irradiation, a strain field is introduced by the defects, which can distort the original lattice. A large lattice parameter in itself is not better or worse than a small lattice parameter, but the change in the lattice parameter (compared to the original sample) is undesirable and this change is caused by accumulation of defects (i.e., radiation damage). Lattice expansion (strain) is commonly used to evaluate radiation damage.

Comment 4:

The authors added a comment that partial melting might have occurred during synthesis. While that can't be ruled out, it seems unlikely based on the SEM observations as it should be possible to see evidence of any incipient melting in the microstructure. The authors might want to reconsider using this explanation.

Thank you for your comments. We agree with the reviewer that no features of incipient melting can be observed in the original Fig. S9. We have now included the SEM observation of incipient melting in HEC-Mo as an example (Fig. S11c in the revised supplementary information). We find that there are several examples of the formation of pockets at grain boundary junctions that are characteristic of the presence of molten zones as marked by arrows (Fig. S11c). We have included this discussion in our revised manuscript.

Comment 5:

I guess a final note is that it's unclear to me why the authors focused on the Mo-HEC instead of the Zr-HEC given their conclusion that the Zr-HEC has a greater tendency for CSRO. I would have thought that they would have conducted the heat treatments to influence "disordering" and to follow the irradiation damage on the Zr-HEC.

Thank you for your comments. We didn't perform annealing on HEC-Zr sample as well as the following irradiation because this work is a fundamental study and we are just selecting the HEC-Mo as a representative. Thermal annealing on HEC-Zr and the following irradiation is beyond the scope of this work.

Point by point response to reviewer's comments

Reviewer #1:

After carefully re-evaluating the manuscript and the supporting information, I have two remaining points:

Comment 1:

First, the central narrative of the paper relies heavily on the claim that the authors have two distinct experimental states for the Mo-containing carbide: one with CSRO and one without CSRO. This distinction is essential for their interpretation of the radiation-damage results as well as for the broader implications they draw about tunability of CSRO. However, the manuscript does not present any experimental evidence that actually confirms the existence of a “zero-CSRO” state. All experimental studies I am aware of report that increasing temperature reduces CSRO but does not eliminate it entirely; in practice, full randomization is extremely difficult to achieve (if not impossible). Furthermore, there are also studies that show that even a very rapid cooling cannot avoid the CSRO formation during cooling at high temperatures. In the present manuscript, the calorimetry data seems to demonstrate that one processing condition has more CSRO than the other. This is not the same as demonstrating that the higher-temperature condition has no CSRO. On the contrary, based on the literature and on the authors' own data, I would expect measurable (though reduced) CSRO to remain. Quantifying this difference may indeed be challenging, and I agree that a qualitative comparison is still valuable, but presenting the lower-CSRO state as “fully disordered” is not supported by the data as written.

Thank you for pointing this out. We agree with the reviewer that it is very difficult or even impossible to reach a fully disordered state. Though it is impossible to eliminate CSRO from HECs entirely and it is difficult to quantify the difference in the extent CSRO between as-synthesized HEC-Mo and HEC-Mo annealed at 1773 K, qualitatively, our data provides strong evidence that CSRO has been significantly diminished in HEC-Mo annealed at 1773 K. Thus, our claims regarding the role of CSRO (e.g., that CSRO can suppress radiation-induced damage in HECs) remain robust. We have modified our manuscript to refer HEC-Mo annealed at 1773 K as HEC-Mo with low CSRO and as-synthesized HEC-Mo or HEC-Mo annealed at 1573 K as HEC-Mo with high CSRO.

Comment 2:

Second, regarding Figure S7, I do not see the evidence the authors claim to be present. If I understand the intention correctly, the authors are arguing that adding CSRO increases the diffuse background intensity in the CBED pattern, and that this increase can be used as an indicator of CSRO. However, many other factors produce the same effect, including specimen thickness, temperature and local crystal distortions unrelated to chemical order. CSRO is typically identified through specific diffuse-scattering features—extra maxima at particular reciprocal-space positions that correlate with ordering vectors—not via a general increase in background. Even the use of diffuse scattering for CSRO identification is non-trivial, as several papers have pointed out challenges in distinguishing CSRO-related diffuse features from other intrinsic and extrinsic contributions. Given these considerations, I do not find that Figure S7 meaningfully supports the manuscript's claims. The figure does not provide clear or interpretable evidence of CSRO, and, as currently presented, does not advance the narrative.

Thank you for your comments. We agree with the reviewer that many factors can produce the same diffuse scattering effects as what we have observed in Fig. S7, though we have tried our best to control such factors, including sample thickness. Given that this figure is not essential to support our conclusions, we have removed it from our revised Supplementary information and no longer refer to it in our revised main manuscript.

Reviewer #3:

I have reviewed the revised paper on high entropy carbides and feel that the authors have adequately addressed my concerns, which were similar to the concerns of the other reviewers. However, their modifications A few minor comments:

Comment 1:

The authors incorrectly refer to an order-disorder transition in their discussion of their thermal analysis data. The classic description of order-disorder transitions involves going from a fully disordered phase to a fully long-range ordered phase either by a first-order or a second-order transition. In the HEC materials, the structure is already ordered but with the possibility that the atoms on the cation sites are not completely random and undergo CSRO. Thus, the authors should modify their description accordingly, i.e., they should refrain from referring to their CSRO structure as resulting from an ordering transition given the structure does not change crystal structure.

Thank you for bringing our attention to this point. We agree with the reviewer that referring to the dissolution of CSRO as order-to-disorder transition can be confusing or inaccurate because it does not involve changes in the structural order. We have modified our manuscript and supplementary information accordingly. Instead, the process is referred to as reduction of CSRO with the increase of temperature.

Comment 2:

The authors refer to Fig. S11c in the text to represent partial (incipient) melting – it should be Fig. S12c.

Thank you for your comments. We have read through our manuscript to correct typos and mistakenly referred images.